# Histone deacetylase 1 maintains lineage integrity through histone acetylome refinement during early embryogenesis

Jeff Jiajing Zhou[1], Jin Sun Cho[1], Han Han[1], Ira L Blitz[1], Wenqi Wang[1], Ken WY Cho[1,2]*

[1]Department of Developmental and Cell Biology, University of California, Irvine, Irvine, United States; [2]Center for Complex Biological Systems, University of California, Irvine, Irvine, United States

**Abstract** Histone acetylation is a pivotal epigenetic modification that controls chromatin structure and regulates gene expression. It plays an essential role in modulating zygotic transcription and cell lineage specification of developing embryos. While the outcomes of many inductive signals have been described to require enzymatic activities of histone acetyltransferases and deacetylases (HDACs), the mechanisms by which HDACs confine the utilization of the zygotic genome remain to be elucidated. Here, we show that histone deacetylase 1 (Hdac1) progressively binds to the zygotic genome from mid-blastula and onward. The recruitment of Hdac1 to the genome at blastula is instructed maternally. *Cis*-regulatory modules (CRMs) bound by Hdac1 possess epigenetic signatures underlying distinct functions. We highlight a dual function model of Hdac1 where Hdac1 not only represses gene expression by sustaining a histone hypoacetylation state on inactive chromatin, but also maintains gene expression through participating in dynamic histone acetylation–deacetylation cycles on active chromatin. As a result, Hdac1 maintains differential histone acetylation states of bound CRMs between different germ layers and reinforces the transcriptional program underlying cell lineage identities, both in time and space. Taken together, our study reveals a comprehensive role for Hdac1 during early vertebrate embryogenesis.

*For correspondence:
kwcho@uci.edu

Competing interest: The authors declare that no competing interests exist.

## Editor's evaluation

In this revised and important study, the authors investigate the roles of histone deacetylases in the spatial epigenetic regulation of zygotic gene expression during embryo gastrulation. They provide convincing evidence for HDAC1 binding to genes around the timing of large-scale genome activation, and that inhibition of histone acetylation blocks gastrulation, blurring cell lineage integrity, tied to both positive and negative regulatory effects on transcription in space and time. The research reveals new insight on the role of histone acetylation-deacetylation in dynamics in epigenetic control of gene expression and cell fate determination during early tissue patterning in embryogenesis.

## Introduction

A fundamental question in early development is the mechanism of zygotic genome activation (ZGA), which requires the degradation of maternal mRNAs and the activation of embryonic transcription (*Blitz and Cho, 2021*, *Tadros and Lipshitz, 2009*). During ZGA, the embryonic genome undergoes a dramatic reprogramming of gene expression, which is also accompanied by remodeling of the embryonic epigenome. Post-translational modifications to histones are a major epigenetic regulation influencing chromatin structure and thus play a central role in ZGA. Histone acetylation appears during the

onset of both minor and major ZGA waves in many species. In *Drosophila*, histone acetylation occurs at mitotic cycle 8 on a few early zygotic genes (*Li et al., 2014*). miR430, the first zygotically active gene, is marked by H3K27ac in 64-cell staged zebrafish embryos (*Chan et al., 2019*). Genome-wide H3K27ac is detected at mid-blastula shortly after the onset of ZGA in *Xenopus* (*Gupta et al., 2014*). In mice, the zygotic genome is increasingly marked by H3K27ac from immature and metaphase II oocytes to 2-cell-stage embryos (*Dahl et al., 2016*). Despite these findings, there remain several major questions. How is the interplay of enzymes regulating histone acetylation deployed in developing embryos? What is the role of observed histone acetylation on gene expression? How are the spatial and temporal patterns of histone acetylation established during ZGA?

Histone acetylation occurs on the ε-amino group of the lysine residues within N-terminal tails of all four core histones (*Inoue and Fujimoto, 1969*; *Seto and Yoshida, 2014*). Acetylation is a reversible process that is directly catalyzed by opposing activities of histone acetyltransferases (HATs) and histone deacetylases (HDACs). In addition, HATs and HDACs can also regulate the acetylation of lysine residues on non-histone proteins (*Choudhary et al., 2009*). Histone acetylation is often associated with active gene transcription because the acyl groups neutralize the positive charge on the lysine residues, thereby reducing the affinity of histones to DNA (*Wang et al., 2000*; *Anderson et al., 2001*); it also serves as a binding platform for bromodomain (BRD) proteins which scaffold and stimulate the transcriptional machinery (*Hassan et al., 2007*; *Filippakopoulos et al., 2012*). The balance between HATs and HDACs directly shapes histone acetylation landscapes and subsequently affects transcriptomes.

HDACs are critical epigenetic regulators because they reset chromatin states by returning acetylated lysine residues on histones to the basal state, which can subsequently be subjected to alternative modifications such as methylation. HDACs are grouped into four classes based on phylogenetic conservation. Class I (HDAC1, 2, 3, 8), Class II (HDAC4, 5, 6, 7, 9, 10), and Class IV (HDAC11) HDACs are zinc dependent and are related to yeast Rpd3, Had1, and Hos3, respectively; Class III (SIRT1, 2, 3, 4, 5, 6, 7) HDACs, also known as Sirtuins, are NAD$^+$ dependent and are related to yeast Sir2 (*Gregoretti et al., 2004*; *Milazzo et al., 2020*). HDACs are well-characterized negative regulators of gene expression during development. For example, Hdac1 silences homeotic genes in cooperation with Polycomb group repressors in *Drosophila* (*Chang et al., 2001*). In zebrafish, Hdac1 represses Notch targets during neurogenesis (*Cunliffe, 2004*; *Yamaguchi et al., 2005*). In *Xenopus*, HDAC activity suppresses Vegt-induced ectopic mesoderm in ectoderm lineages (*Gao et al., 2016*), represses multi-lineage marker genes at blastula (*Rao and LaBonne, 2018*), and desensitizes dorsal Wnt signaling at late blastula (*Esmaeili et al., 2020*). Conversely, HDACs can also positively regulate gene expression. For instance, inhibition of HDAC activities rapidly down-regulates some genes in yeast, suggesting an activator function of HDACs (*Bernstein et al., 2000*). Genetic deletions or pharmacological application of HDAC inhibitors in cell lines results in both up- and down-regulation of genes (*Reid et al., 2005*; *Zupkovitz et al., 2006*; *Meganathan et al., 2015*). Furthermore, genome-wide studies showed that HDACs occupy genomic loci of active genes, and their binding correlates with gene activities (*Kurdistani et al., 2002*; *Wang et al., 2002*; *Wang et al., 2009*; *Kidder and Palmer, 2012*). These seemingly opposing functions of HDACs raise an important question as to the exact roles of HDACs on chromatin states and transcriptomes in developing embryos.

In this study, we focus on the role of Hdac1 in regulating the zygotic epigenome and transcriptome during *Xenopus* germ-layer specification coinciding with ZGA. Current evidence in *Xenopus* as well as in other non-mammalian systems suggests that the early embryonic genome is rather naive and that major chromatin modifications occur at or after ZGA (*Bonn et al., 2012*; *Vastenhouw et al., 2010*; *Gupta et al., 2014*; *van Heeringen et al., 2014*; *Hontelez et al., 2015*). Thus, the system allows us to probe the earliest establishment of histone acetylation and dissect the link between actions of Hdac1, the zygotic histone acetylome, and zygotic transcriptome during the first cell lineage segregation event. Here, we report that the major Hdac1 binding to the embryonic genome occurs at blastula; the binding of Hdac1 during this stage is dependent on maternal factors. We highlight a dual function model for Hdac1. First, Hdac1 keeps inactive chromatin free of histone acetylation, preventing gene misactivation in respective germ layers. Second, Hdac1 participates in dynamic histone acetylation–deacetylation cycles on active chromatin, sustaining the expression of genes that are enriched in respective germ layers. Taken together, our study reveals a coordinated spatial and temporal regulation by Hdac1 during ZGA.

## Results

### Hdac1 binds to the genome progressively during blastula and onward

To identify the major functional candidates of HDACs during the early *Xenopus* embryogenesis, we examined the temporal RNA expression profiles (*Owens et al., 2016*) of HDAC family members (Class I, II, and III HDACs) from the zygote to the beginning of the neurula stage in *Xenopus tropicalis*. The RNA expression level of *hdac1* is the highest among all HDACs examined, followed by *hdac2* (*Figure 1—figure supplement 1A*). Both Hdac1 and Hdac2 proteins are present in the unfertilized egg to the mid gastrula stage, and the overall expression levels of Hdac1 and Hdac2 are relatively constant during this period (*Figure 1A*). These data reveal that Hdac1/2 are the major maternally endowed HDACs functioning during this window of development.

Since Hdac1 modulates various aspects of transcriptional regulation and the chromatin landscape, we examined genome-wide Hdac1-binding patterns during early germ-layer development (*Figure 1—figure supplement 1B*) using chromatin immunoprecipitation (ChIP) assays followed by sequencing (ChIP-seq). A set of high-confidence peaks at each stage were obtained using irreproducibility discovery rate (IDR) analysis (*Li et al., 2011*) from two biologically independent samples (*Figure 1—figure supplement 1C*). Hdac1 binds to 1340 regions at the mid-blastula (st8), 5136 regions at the late blastula (st9), and 22,681 regions at the early gastrula (st10.5) stages (*Figure 1B*). Overall, a minority of Hdac1 peaks are unique to each of the blastula stages (Clusters a and b) and a majority of Hdac1 peaks are present across multiple stages (Cluster c) and at the early gastrula stage (Cluster d) (*Figure 1C*, *Figure 1—figure supplement 1D*). The expression levels of zygotic genes associated with Cluster c Hdac1 peaks are higher than those associated with Clusters a, b, and d, suggesting that the genes associated with sustained Hdac1 binding are more active during gastrulation (*Figure 1D*). Similarly, Hdac2 gradually accumulates on the embryonic genome (*Figure 1—figure supplement 1E*). Interestingly, 99% of Hdac1 peaks at stage 9 and 97% of Hdac1 peaks at stage 10.5 overlap with Hdac2 peaks (*Figure 1—figure supplement 1F, G*). When the levels of Hdac2 peaks associated with Hdac1 were compared to Hdac2 peaks , Hdac1 peaks display a significantly higher Hdac2 peak enrichment (*Figure 1—figure supplement 1H*). Together, we revealed that Hdac1/2 are progressively directed to the genome during early development. Since a majority of Hdac1-bound peaks are similarly bound by Hdac2, we focus on Hdac1 peaks in subsequent analyses.

To investigate the differences of Hdac1 genomic occupancy across early stages, we examined various genomic features of regions bound by Hdac1 at each stage. The majority of Hdac1 peaks are found in either intergenic or intronic regions; a minor fraction of Hdac1 peaks is present within exons or transcriptional termination sites. A notable observation is the increased Hdac1 binding to more promoter regions over developmental times (*Figure 1E*). We then analyzed the distribution of Hdac1 across three stages along the gene bodies of genes bound by Hdac1. A higher enrichment of Hdac1 binding is located near the promoters of genes as development proceeds (*Figure 1F*). Importantly, the timing of Hdac1-binding accumulation coincides with the mid-blastula stage (st8) (*Figure 1G*), suggesting that Hdac1 is involved in the epigenetic regulation of these genes during ZGA.

### Blastula Hdac1 binding is maternally instructed

Since Hdac1 binds progressively to the genome during ZGA, we investigated the contribution of zygotic factors to the recruitment of Hdac1 at the late blastula stage. We injected α-amanitin, which blocks both transcription initiation and elongation (*Chafin et al., 1995*), to block embryonic transcription (*Figure 2—figure supplement 1A*). A high Pearson correlation between α-amanitin-injected and uninjected embryos is observed genome-wide or at all Hdac1 IDR peaks (*Figure 2A*), which is higher than the Pearson correlation representing the batch effect when compared to a different batch of same staged embryos (WT) (*Figure 2—figure supplement 1B*). There is no significant signal difference between α-amanitin-injected and uninjected embryos at all Hdac1 IDR peaks (*Figure 2—figure supplement 1C, H*). We conclude that Hdac1 binding is independent of zygotic transcription at the late blastula stage, suggesting the importance of maternal factors in Hdac1 recruitment.

To identify maternal factors that facilitate the recruitment of Hdac1 to the genome, we performed a de novo motif search of the DNA sequences of 5136 st9 Hdac1peaks. Sox, Foxh1, and Pou motifs are found to be the most frequent maternal TF motifs (*Figure 2B*, *Supplementary file 1*). We thus compared the genomic binding profiles of Hdac1 to two maternal TFs, Foxh1 (*Charney et al., 2017*) and Sox3. A majority of Hdac1-bound regions (Cluster 1) overlaps with both Foxh1- and Sox3-bound

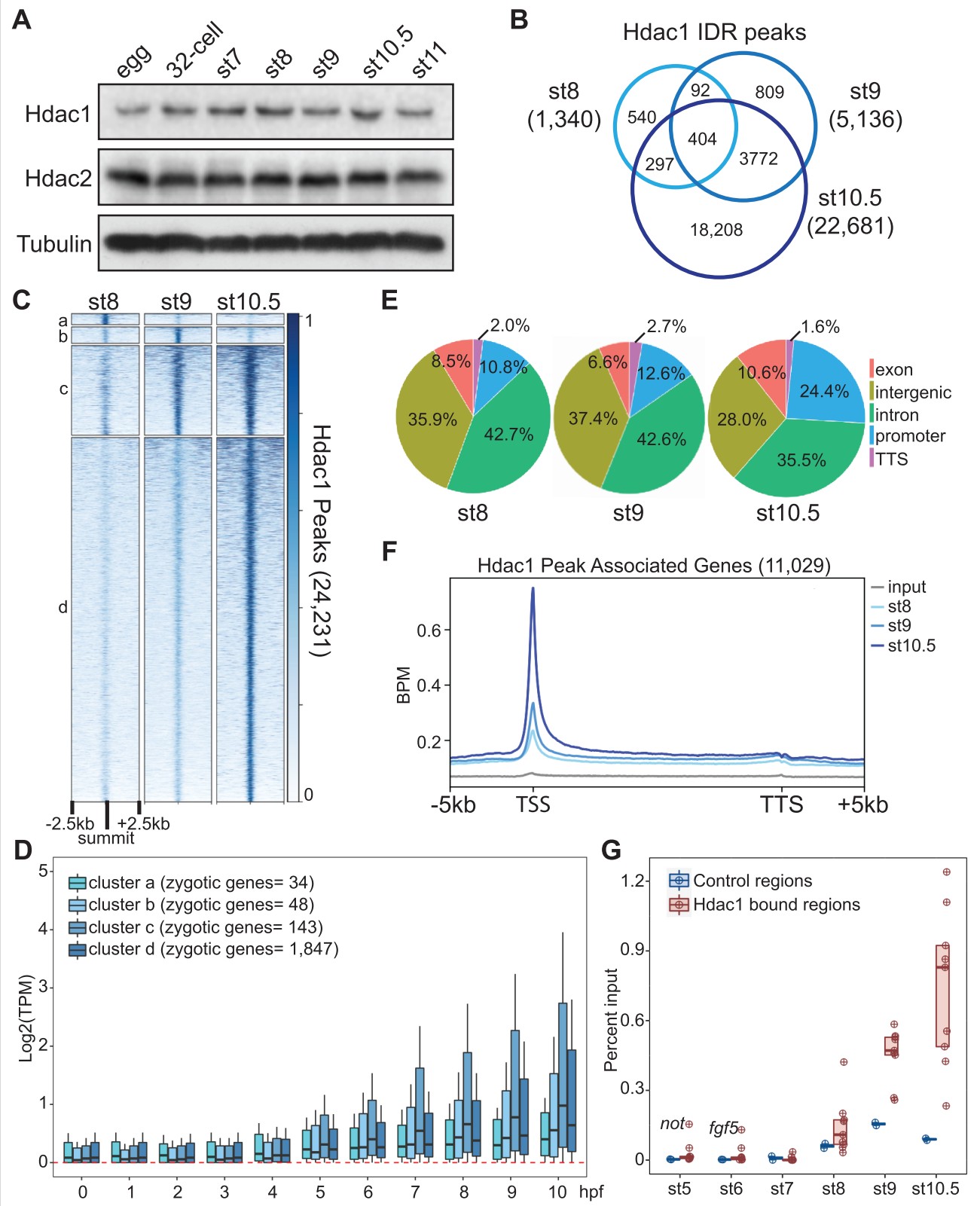

**Figure 1.** Hdac1 binds to the genome gradually during early *Xenopus* development. (**A**) Western blot analyses showing Hdac1 and Hdac2 proteins over a time course of early development. β-Tubulin is used as a loading control. (**B**) Venn diagram comparing Hdac1 irreproducibility discovery rate (IDR) peaks among examined stages. The sums of peaks in st8 and st9 are smaller due to instances where more than one peak from st8 or st9 overlaps the same single st10.5 peak. (**C**) Clustered heatmap showing Hdac1 ChIP-seq signals at each stage over a window of 5 kb centered on the summit of

*Figure 1 continued on next page*

*Figure 1 continued*

all Hdac1 IDR peaks in descending order. (**D**) The expression levels (TPM, transcripts per million) of genes associated with Hdac1 clusters at different developmental periods. (**E**) Distributions of Hdac1 IDR peaks at each stage across five defined genomic features. The promoter is defined as −1 kb to +100 bp from TSS (transcription start site) while the TTS (transcription termination site) is defined as −100 bp to +1 kb from TTS. (**F**) Distributions of Hdac1 ChIP-seq signals within the intervals of 5 kb upstream of gene model 5′ ends, gene bodies (normalized for length), and 5 kb downstream of gene model 3′ ends at each stage. The signal of st9 input DNA ChIP-seq is used as a negative control. Y-axis values represent reads quantified by bins per million (BPM) at a bin size of 50 bp. (**G**) ChIP-qPCR showing Hdac1 enrichment on selected genomic regions (nine positive regions: *alkbh2*, *fgf5*, *foxi4.2*, *gdnf*, *hhex*, *miR428a*, *not*, *snai1*, and *sp8*; two negative regions: *hspa4* and *klf11*) at indicated stages of early embryogenesis.

The online version of this article includes the following source data and figure supplement(s) for figure 1:

**Source data 1.** Western blot analyses showing Hdac1 and Hdac2 proteins during *Xenopus embryogenesis*.

**Figure supplement 1.** Hdac1 genomic occupancy in early *Xenopus* development.

regions while only a small fraction of Hdac1-bound regions (Cluster 2) overlaps the binding of either Foxh1, or Sox3, or neither (*Figure 2C*). More than 80% of Hdac1 peaks overlap with Foxh1 or Sox3 peaks (*Figure 2—figure supplement 1D, E*). A positive correlation between Hdac1 binding and Foxh1/Sox3 binding is observed at all Hdac1 IDR peaks (*Figure 2—figure supplement 1F, G*). We noted frequent overlapping binding of Hdac1 with Foxh1/Sox3 (*Figure 2—figure supplement 1I, J*), and highly enriched signals of Hdac1, Foxh1, and Sox3 present around promoters of genes (*Figure 2D*). All these observations suggest a role for Foxh1 and Sox3 in Hdac1 recruitment. We confirmed the co-occupancy of Hdac1 with each of Foxh1 and Sox3 TFs on the same DNA molecules using sequential ChIP-qPCR (*Figure 2E*, *Figure 2—figure supplement 1K, L*). Depletion of Foxh1, Sox3, or both TFs by morpholino injections showed a reduced binding of Hdac1 around Foxh1/Sox3 co-occupied genomic regions (*Figure 2F*). Hence, we propose that Foxh1 and Sox3 maternal TFs facilitate Hdac1 recruitment during ZGA.

## Hdac1 binds to genomic regions with distinct epigenetic signatures

To further characterize regions bound by Hdac1 across early germ-layer development, we examined epigenetic signatures (*Gupta et al., 2014*; *Hontelez et al., 2015*; *Charney et al., 2017*) on Hdac1 peaks across various stages. Ep300 binding (a HAT), which catalyzes the acetylation of histone, is observed in Hdac1 peaks (Clusters b–d) from the late blastula and onward where RNA polymerase II signals also emerge (*Figure 3—figure supplement 1A*). This indicates that Hdac1 and Ep300 share similar binding profiles on many transcriptionally active genes. We next surveyed several histone acetylation modifications in regions bound by Hdac1 (*Figure 3A*). Consistent with the overlapping binding of Ep300, Hdac1 peaks display signals of H3K9ac (Clusters c and d), H3K18ac (Clusters b–d), H3K27ac (Clusters b–d), and pan-H3 lysine acetylation (pan-H3Kac) (Clusters b–d). We then examined several histone methylation modifications that are associated with gene activation (*Figure 3—figure supplement 1B*). H3K4me1, a primed enhancer mark (*Creyghton et al., 2010*), and H3K4me3, an active promoter mark (*Heintzman et al., 2007*), display signals at Hdac1-bound regions (Clusters c and d); H3K36me3, a transcription elongation mark (*Kolasinska-Zwierz et al., 2009*), displays minimal signals at any Hdac1-bound regions. These observations reveal that Hdac1 binds to genomic regions with active epigenetic signatures.

Hdac1 resets the state of acetylated histones by removing acetyl groups, and thus can facilitate the formation of repressive chromatin. Here, we analyzed several histone methylation modifications associated with gene repression (*Figure 3B*). Hdac1 binds to genomic regions mostly free of H3K9me2, H3K9me3, or H4K20me3. All three modifications are known to mark constitutive heterochromatin denoting gene-poor areas consisting of tandem repeats (*Richards and Elgin, 2002*). In addition, a fraction of Hdac1-bound regions (Clusters c and d) display signals of H3K27me3 (*Figure 3B*), which is known to mark facultative heterochromatin consisting of developmental-cue silenced genes (*Trojer and Reinberg, 2007*). These observations suggest that Hdac1 binds to facultative heterochromatic regions facilitating the repression of genes.

Since the majority of Hdac1 peaks (Clusters b–d) are marked by both active and repressive epigenetic signatures, we wonder how Hdac1 functions in epigenetically distinct genomic loci. We compared peaks between two functionally opposing histone modifications H3K27ac and H3K27me3 to Hdac1 peaks, then subdivided these Hdac1 peaks into four clusters representing functionally distinct CRM types (*Figure 3C*, *Figure 3—figure supplement 1C*). Cluster I denotes 3548 Hdac1 peaks marked by

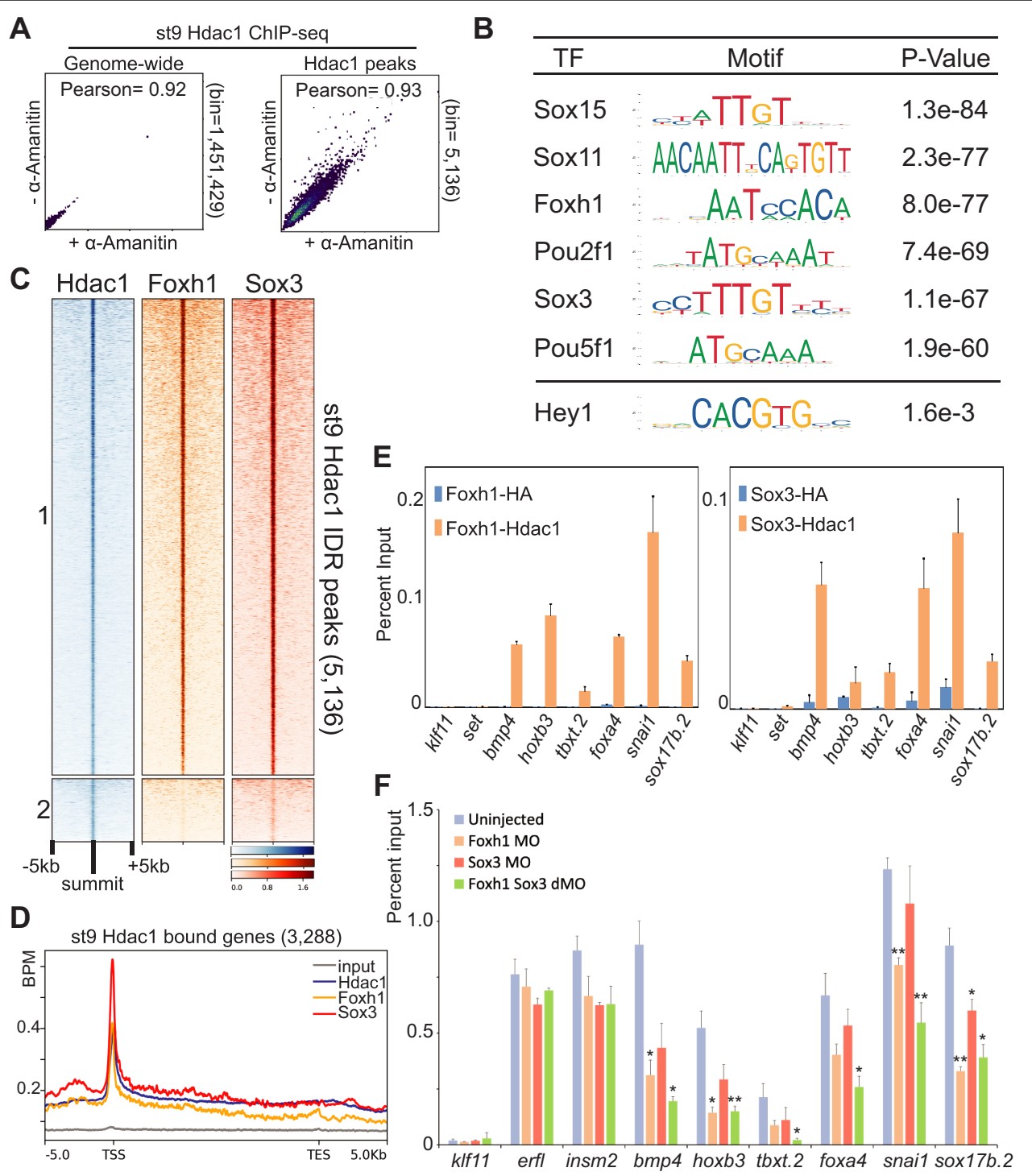

**Figure 2.** Maternal factors instruct Hdac1 recruitment during blastula stages. (**A**) Pairwise Pearson correlation analyses comparing ChIP-seq signals of st9 Hdac1 irreproducibility discovery rate (IDR) peaks between α-amanitin-injected and uninjected embryos across the genome (left) and among Hdac1 IDR peaks (right). (**B**) Motif analyses of st9 Hdac1 peaks (500 bp centered on IDR peak summit). Motif sequence to the corresponding factor is retrieved from JASPAR. Hey1 is an example of TF motif with low significance. (**C**) Clustered heatmap depicting st9 Foxh1 and Sox3 ChIP-seq signals in a window of 10 kb centered on st9 Hdac1 IDR peaks with descending order. (**D**) ChIP-seq signal enrichment of Hdac1, Foxh1, and Sox3 within the intervals of 5 kb upstream of gene model 5' ends, gene bodies (normalized for length), and 5 kb downstream of gene model 3' ends of st9 Hdac1 associated genes. The signal of st9 input DNA ChIP-seq is used as a negative control. Y-axis values represent reads quantified by bins per million (BPM) at a bin size of 50 bp. (**E**) St9 sequential ChIP-qPCR analyses for Foxh1 and Hdac1 co-bound regions and Sox3 and Hdac1 co-bound regions. anti-HA is used as a negative

*Figure 2 continued on next page*

*Figure 2 continued*

control. The error bars represent the variation from two technical replicates. (**F**) ChIP-qPCR analysis of Hdac1 peaks that are also Foxh1/Sox3 co-bound after Foxh1 and/or Sox3 depletion. Genomic loci associated with *klf11* (no Hdac1 signals), *erfl* and *insm2* (Hdac1 signals without Foxh1 or Sox3 signals) are negative controls. The error bars represent the variation from two technical replicates. Student's *t*-test is used to calculate p-values over Hdac1 enrichment of uninjected embryos. * is for p< 0.05** is for p< 0.01.

The online version of this article includes the following figure supplement(s) for figure 2:

**Figure supplement 1.** Hdac1 binding to genome is instructed maternally.

both H3K27ac and H3K27me3 (*Figure 3D*). Given that both H3K27ac and H3K27me3 are modified on the same lysine residue, we speculate that these regions are differentially marked in space due to heterogeneous cell populations present in the whole embryo. Therefore, Hdac1 Cluster I peaks are referred to as heterogeneous CRMs. Cluster II denotes 1389 Hdac1 peaks marked by only H3K27me3 indicating that these regions are associated with inactive developmental genes (*Figure 3D*). Hdac1 Cluster II peaks represent repressive CRMs. Cluster III denotes 13,669 Hdac1 peaks marked by only H3K27ac, suggesting that these are active CRMs. Cluster IV denotes 4836 Hdac1 peaks with neither H3K27ac nor H3K27me3 modifications (*Figure 3D*). At genomic loci marked with two distinct H3K27 modifications (*Figure 3—figure supplement 1D*), the expression levels of genes bound by Hdac1 (I, II, and III) are generally higher than that of genes unbound (I', II', and III') (*Figure 3—figure supplement 1E*), suggesting that Hdac1 binding correlates with transcriptional activity of the genes. Together, we show that Hdac1-bound CRMs are subject to distinct epigenetic modifications, which confer differential CRM activities.

## HDAC activity is required for differential germ-layer histone acetylomes

A major function of HDACs is to catalyze the removal of acetyl groups from histones. We hypothesize that Hdac1 differentially regulates histone acetylation of four different Hdac1 CRM Clusters (Clusters I–IV in *Figure 3C*). To test this hypothesis, we treated embryos continuously with a widely used pan-HDAC inhibitor, Trichostatin A (TSA) (*Yoshida et al., 1990*) beginning at the 4-cell stage and followed the development up to tailbud stages. Embryos treated with TSA are developmentally arrested at gastrula (*Figure 4A*). We observed the presence of dorsal blastopore lip, suggesting that the progression but not the initiation of gastrulation is defective. A Class I HDAC inhibitor, valproic acid (VPA) (*Göttlicher et al., 2001*), also produces a similar phenotype (*Figure 4—figure supplement 1A*). We first showed that protein levels of Hdac1 and Hdac2 are not affected by TSA treatment (*Figure 4—figure supplement 1B*). Next, Hdac1 ChIP-seq experiments display high Pearson correlations on embryos treated with solvent control or TSA at st9 (*Figure 4—figure supplement 1C*) and st10.5 (*Figure 4—figure supplement 1D*), which is further supported by higher Pearson correlations upon TSA treatment than the batch effect (*Figure 4—figure supplement 1E, F*). Lastly, differential peak analysis on Hdac1 IDR peaks in embryos treated with solvent control or TSA showed barely any differential Hdac1 signals (0.1% peaks or less) at st9 (*Figure 4—figure supplement 1G*) and st10.5 (*Figure 4—figure supplement 1H*). These results suggest that TSA treatment of early *Xenopus* embryos does not alter the recruitment of Hdac1 to the genome. To examine the efficacy of HDAC activity inhibition by TSA, we surveyed six well-known histone acetylation modifications by western blot. Drastically increased levels of H3K9ac, H3K18ac, and H3K27ac are observed in TSA-treated embryos (*Figure 4B*; *Rao and LaBonne, 2018*), whereas H3K14ac, H3K56ac, and H4K20ac are not detected during this stage of development. These data indicate that HDAC activity is required to maintain the proper level of histone acetylation during gastrulation.

Given that Hdac1 CRM Clusters (Clusters I–IV in *Figure 3C*) are subjected to both active and repressive epigenetic modifications presumably in different germ layers, we compared the general status of the H3 acetylome (pan-H3Kac) between two distinct germ layers, the animal cap (AC, presumptive ectoderm) and the vegetal mass (VG, presumptive endoderm). We performed pan-H3Kac ChIP-seq at early gastrula (st10.5) on dissected AC and VG explants from embryos treated with either TSA or solvent control (*Figure 4—figure supplement 2A*). A comparison of high-confidence peaks from each explant reveals that a majority (~75%) of pan-H3Kac are specifically marked in ectodermal (~37%) and endodermal (~38%) germ layers (*Figure 4C*, *Figure 4—figure supplement 2B*). We observed the

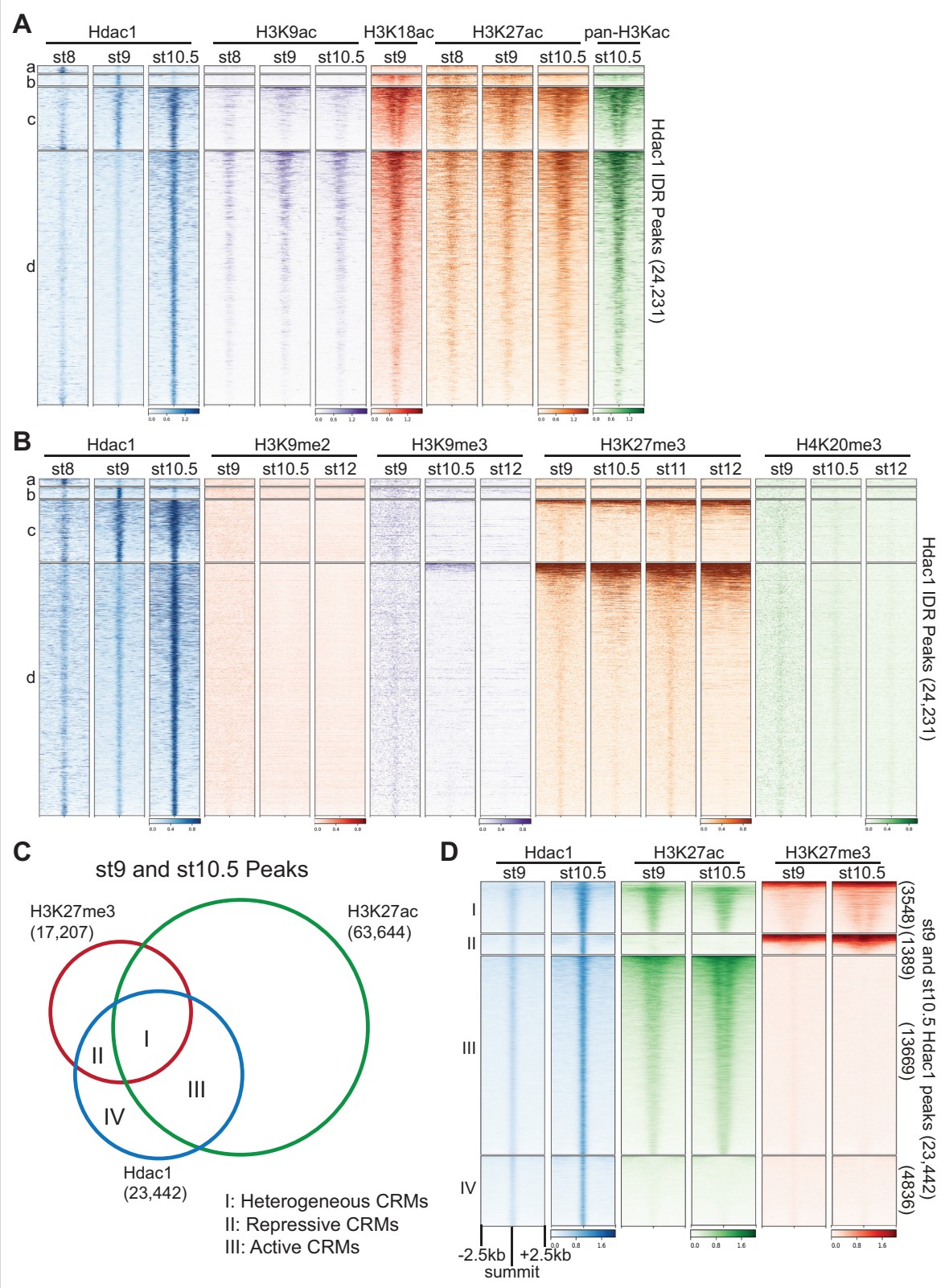

**Figure 3.** Hdac1 binds to *cis*-regulatory modules (CRMs) containing functionally distinct histone modifications. Clustered heatmaps showing signals from several stages of various (**A**) histone acetylation marks and (**B**) repressive histone methylation marks on Hdac1 peaks. Each cluster corresponds to the same regions present in *Figure 1C*. (**C**) Venn diagram illustrating Hdac1 peaks overlapping with H3K27me3 and H3K27ac peaks from both st9 and st10.5 combined. (**D**) Clustered heatmaps depicting signals of H3K27me3 and H3K27ac on combined st9 and st10.5 Hdac1 peaks. Clusters denote the

*Figure 3 continued on next page*

*Figure 3 continued*

same genomic regions in **C**. Numbers on the right side indicate the total number of regions in each cluster. The signals are shown in a window of 5 kb centered on the summits of Hdac1 peaks presented in descending order of track signal intensities within each cluster.

The online version of this article includes the following figure supplement(s) for figure 3:

**Figure supplement 1.** Hdac1-bound *cis*-regulatory modules (CRMs) are marked with distinct epigenetic signatures.

signal intensity of pan-H3Kac in AC/VG shared genomic regions (Cluster A in *Figure 4C*) is significantly higher than those of AC- or VG-specific genomic regions (Clusters B and C in *Figure 4C*), which also coincides with higher gene expression levels (*Figure 4—figure supplement 2C*). To correlate the differential pan-H3Kac states and the differential gene expression profiles between the two germ layers, we assigned high enrichment pan-H3Kac peaks (~top 30%) to the nearest genes within 10 kb and compared the expression levels of these genes in each germ layer (*Blitz et al., 2017*). Indeed, the expression level of genes enriched with AC-specific pan-H3Kac are higher in AC than VG (*Figure 4—figure supplement 2D*), and the expression levels of genes enriched with VG-specific pan-H3Kac are higher in VG than AC (*Figure 4—figure supplement 2E*). Well-known genes with germ-layer-specific expression exhibit localized pan-H3Kac signals between germ layers (*Figure 4—figure supplement 2F*). These results illustrate that the histone acetylation profile generally coincides with the animally and vegetally localized expression of transcripts.

To uncover the role of Hdac1 in regulating histone acetylation states of CRM clusters (Clusters I–IV in *Figure 3D*), we first examined the general distribution of pan-H3Kac in these Hdac1 CRM Clusters. Consistent with H3K27 modifications, both heterogeneous CRMs (Cluster I, both H3K27ac and H3K27me3) and active CRMs (Cluster III, only H3K27ac), but not repressive CRMs (Cluster II, only H3K27me3), are marked by pan-H3Kac. Nearly half of pan-H3Kac marks on heterogeneous or active CRMs are localized either animally or vegetally (*Figure 4C, D*), suggesting that these CRMs are regionally active in specific germ layers. We then quantitatively (*Egan et al., 2016*) compared the levels of pan-H3Kac signals (read density) on each CRM within Hdac1 CRM Clusters I–IV with and without TSA treatment. A global increase of pan-H3Kac signals across all Hdac1 CRM Clusters is observed after TSA treatment (*Figure 4E*, *Figure 4—figure supplement 3A*), which is consistent with the western blot data (*Figure 4B*). TSA-induced HDAC inhibition leads to elevated pan-H3Kac signals on both readily acetylated genomic loci (Cluster α of *Figure 4—figure supplement 3B, C*) and other genomic regions (Cluster β of *Figure 4—figure supplement 3B, C*). Interestingly, we found that CRMs of Hdac1 Clusters I–IV respond differently upon HDAC inhibition: repressive CRMs (Cluster II, only H3K27me3) show the highest fold increase of pan-H3Kac signals, while active CRMs (Cluster III, only H3K27ac) show the lowest fold increase of pan-H3Kac signals when compared to other clusters (*Figure 4F*, *Figure 4—figure supplement 3D*). We also note that the increased amount of pan-H3Kac is very similar across different Hdac1 clusters, irrespective of CRMs being repressive or active CRMs (*Figure 4—figure supplement 3E, F*). This suggests that HDACs catalytic activities are similar whether CRMs are repressive or active. In sum, upon HDAC inhibition, Hdac1-bound repressive CRMs are subject to histone hyperacetylation, while Hdac1-bound active CRMs exhibit a further increase of histone acetylation.

Lastly, we explored how Hdac1 CRM Clusters (Clusters I–IV in *Figure 3C*) differentially respond to HDAC inhibition in specific germ layers. For heterogeneous CRMs (Cluster I, both H3K27ac and H3K27me3), the fold increase of pan-H3Kac signals after TSA treatment is examined (*Figure 4—figure supplement 3G*). Interestingly, CRMs with low levels of histone acetylation tend to be more responsive to HDAC inhibition, than CRMs with high levels of histone acetylation. Next, heterogeneous CRMs were subdivided into three spatial categories, that are pan-H3Kac enriched animally (AC CRMs), vegetally (VG CRMs), and ubiquitous CRMs. The fold changes of pan-H3Kac signals of these CRMs were examined after TSA treatment. Germ-layer-specific AC and VG CRMs show the greater changes upon HDAC inhibition (*Figure 4G*). AC CRMs show an increase in pan-H3Kac signals in VG but not in AC after HDAC inhibition. Similarly, VG CRMs acquire an increase of pan-H3Kac signals in AC but not in VG upon HDAC inhibition. A similar trend is also observed on active CRMs (Cluster III, only H3K27ac), emphasizing germ-layer-specific functions of this CRM cluster (*Figure 4—figure supplement 3H, I*). Taken together, these data demonstrate that Hdac1 activity is spatially regulated in development to maintain proper germ layer specific gene expression patterns.

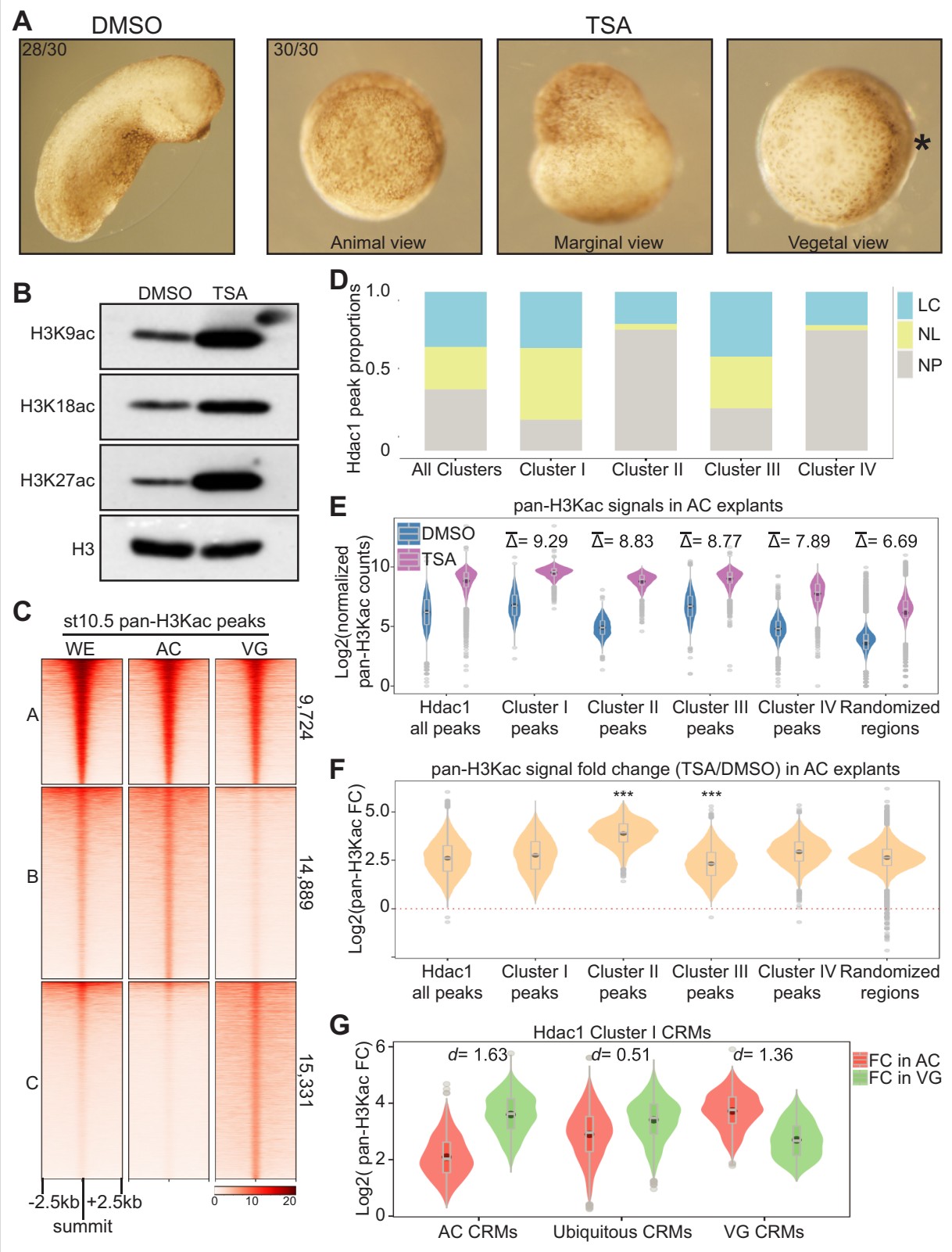

**Figure 4.** Hdac1 maintains differential H3 acetylomes between germ layers. (**A**) Embryos treated with 100 nM Trichostatin A (TSA) displaying gastrulation defects 24 hr post-fertilization. Asterisk denotes the dorsal side containing the early blastopore lip. (**B**) Western blot analyses showing various histone acetylation modifications affected by HDAC inhibition. Anti-H3 is used as a loading control. (**C**) Clustered heatmap depicting signals of pan-H3Kac at st10.5 in whole embryos (WE), animal cap (AC), and vegetal mass (VG) cells. The signals are shown in a window of 5 kb centered on the

*Figure 4 continued*

summits of combined AC and VG peaks presented in descending order within each cluster. (**D**) Stacked bar graph representing proportions of localized (LC) versus non-localized (NL) pan-H3Kac signals found at Hdac1 *cis*-regulatory module (CRM) clusters in *Figure 3C*. A Hdac1 peak is considered to exhibit localized pan-H3Kac if it overlaps with either AC- or VG-specific pan-H3Kac peaks (Cluster B or C in **C**); a Hdac1 peak is considered to exhibit non-localized pan-H3Kac if it overlaps with pan-H3Kac peaks shared between AC and VG (Cluster A in **C**). NP: not overlap with any pan-H3Kac peak. (**E**) Spike-in normalized pan-H3Kac signals across Hdac1 CRM clusters (clusters in *Figure 3C*) in Dimethy Sulfoxide (DMSO)- or TSA-treated AC explants. $\bar{\Delta}$ represents the log2 scaled average differences of spike-in normalized pan-H3Kac signals between DMSO- and TSA-treated AC explants. Randomized genomic regions ($n = 23,442$) are used as the negative control. (**F**) Fold changes (FC) of pan-H3Kac signals at Hdac1 CRM clusters (clusters in *Figure 3C*) in DMSO- or TSA-treated AC explants. Red dotted line denotes the level of zero. *** denotes $p < 0.001$ (Student's *t*-test). (**G**) Fold changes (FC) of pan-H3Kac signals in Cluster I of Hdac1 CRM clusters (clusters in *Figure 3C*) for each spatial CRM category. *d* denotes effect size calculated by Cohen's *d*.

The online version of this article includes the following source data and figure supplement(s) for figure 4:

**Source data 1.** Western blot analyses showing various histone acetylation modifications after HDAC inhibition.

**Figure supplement 1.** Trichostatin A (TSA)-mediated HDAC inhibition does not alter Hdac1 genomic occupancy.

**Figure supplement 1—source data 1.** Western blot analyses of Hdac1 and Hdac2 proteins upon TSA treatment in st9 and st10.5 embryos.

**Figure supplement 2.** Differential H3 acetylomes are established in different germ layers.

**Figure supplement 3.** Germ-layer-specific H3 acetylomes require HDAC activity.

## HDAC activity modulates developmental genes between germ layers

HDACs are considered as transcriptional corepressors because histone deacetylation is generally associated with transcriptional repression. To determine how Hdac1-bound CRM Clusters I–III (*Figure 3C*) influence the activities of their corresponding genes, each CRM within a cluster was assigned to a nearest gene located within 10 kb and genes were placed into one of three classes (*Figure 5A*). Class 1 denotes 3104 genes that have CRMs with mixed marks of H3K27ac and/or H3K27me3 suggesting that these genes are differentially expressed in different germ layers. Class 2 denotes 629 genes whose CRMs are marked with only H3K27me3 indicating that these genes may be repressed. Class 3 denotes 5913 genes whose CRMs are marked with only H3K27ac suggesting that these genes are differentially active in various germ layers. We first investigated the temporal expression pattern (*Owens et al., 2016*) of each gene class. To exclude the interference posed by residual maternal transcripts at these early stages, we examined the expression patterns of exclusively zygotic genes in each class and found that Class 1 and 3 genes are gradually activated after ZGA while Class 2 genes remain mostly silent even at late gastrula (*Figure 5B*). To assess whether Class 2 genes remain inactive throughout development, we extended our temporal expression analysis until tailbud stage 26 (23 hpf). Class 1 and 3 genes are continuously active after ZGA, whereas Class 2 genes are gradually activated from early neurula and onward (*Figure 5C*). These data suggest that Hdac1 regulates both transcriptionally active and silent genes at gastrulation.

To understand how HDAC activity affects the expression of nearby genes, we performed RNA-seq using early gastrula AC or VG explants treated with or without two different Hdac inhibitors, TSA (*Finnin et al., 1999*) and VPA (*Davie, 2003*; *Krämer et al., 2003*). Differential gene expression analyses identified many genes that are affected after TSA or VPA treatment in both AC and VG explants (*Figure 5—figure supplement 1A, C*). Many of the genes affected by TSA treatment were similarly affected by VPA treatment, and vice versa (*Figure 5—figure supplement 1B, C*), suggesting that these are bona-fide HDAC targets. Gene ontology analyses revealed that genes affected by of HDAC inhibition primarily function in early embryonic development such as cell fate commitment, tissue morphogenesis and pattern specification (*Figure 5E*), consistent with the notion that HDACs are important in regulating the genes involved in early embryonic development.

## Integrity of germ-layer-specific transcriptomes is maintained by spatiotemporal HDAC activity

We attempt to correlate how the expression of different classes of Hdac1-bound genes (Classes 1–3 in *Figure 5A*) is affected by HDAC inhibition. We examined the activation of Class 2 genes (Hadc1-bound and H3K27me3 marked) upon HDAC inhibition, which are usually not transcribed until much after gastrulation (*Figure 5B, C*). Interestingly, greater than 85% of Class 2 genes are prematurely activated during gastrulation upon HDAC inhibition (*Figure 5D*), supporting the idea that Hdac1 temporally regulates the expression of Class 2 genes (*Figure 5—figure supplement 1H* ). This is

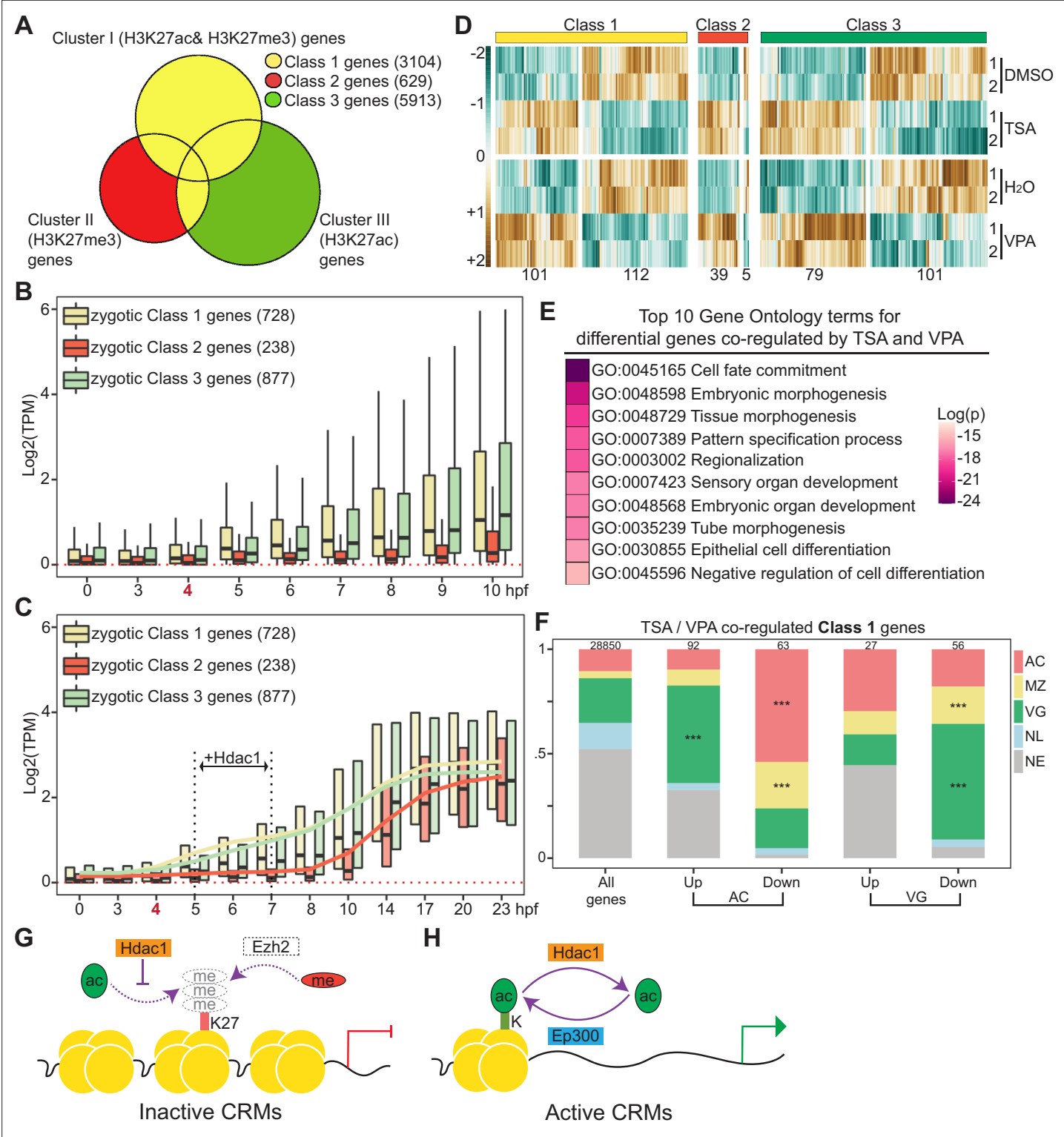

**Figure 5.** Hdac1 regulates germ-layer transcriptomes both in time and space. Venn diagram comparing genes associated with Hdac1 *cis*-regulatory module (CRM) clusters (Clusters I–III in *Figure 3C*). Class 1 genes include genes closest to Cluster I peaks and genes overlapped in Classes 2 and 3; Class 2 are unique genes closest to Cluster II peaks; Class 3 are unique genes closest to Cluster III peaks. (**B**) Time-course TPM expression of zygotic genes in Classes 1–3 from fertilization to 10 hr post-fertilization (hpf, late st12.5). Red dotted line denotes zero. Bold number '4' in red denotes the onset of zygotic genome activation (ZGA). (**C**) Time-course TPM expression of zygotic genes in Classes 1–3 up to 23 hpf (tailbud st26). Black vertical dotted lines denote the time window when Hdac1 binding is examined. Trend lines for each class are generated by connecting mean TPM values at each

*Figure 5 continued on next page*

*Figure 5 continued*

time point. (**D**) The expression profiles of genes affected by Trichostatin A (TSA) or valproic acid (VPA) in each class. The total number of genes in each heatmap cluster is denoted. (**E**) Gene ontology enrichment analysis of genes co-regulated by TSA- and VPA-mediated HDAC inhibition. Only genes with matched gene synonym to *Homo sapiens* are used in the analysis. (**F**) Spatial expression pattern of Class 1 genes co-regulated by TSA and VPA in proportions. The total number of genes in each category is listed at the top of each bar. Only *** denoting p < 0.001 (Fisher's exact test) is shown. AC: animal cap, presumptive ectoderm; MZ: marginal zone, presumptive mesoderm; VG: vegetal mass, presumptive endoderm; NL: non-localized genes; NE: non-expressed genes. (**G, H**) Models of Hdac1 functioning at both inactive and active CRMs: (**G**) on inactive CRMs, H3K27 residue is maintained as unmodified by Hdac1, which may be subjected to H3K27me3-mediated suppression; (**H**) on active CRMs, the state of histone acetylation is dynamically modulated by Ep300- and-Hdac1-mediated acetylation–deacetylation cycles.

The online version of this article includes the following figure supplement(s) for figure 5:

**Figure supplement 1.** Integrity of germ-layer transcriptomes requires HDAC activity.

also consistent with observed histone hyperacetylation at repressive CRMs upon HDAC inhibition (*Figure 4E*, *Figure 4—figure supplement 3A*). Furthermore, we observed that differentially regulated genes in Class 1 (CRMs are marked by a mixture of active H3K27ac and repressive H3K27me3) and Class 3 (whose CRMs are only marked by active H3K27ac) can be up- or down-regulated upon HDAC inhibition (*Figure 5D*). We speculate that HDAC activity Class 1 and 3 genes are spatially regulated.

To gain insights into the spatial gene regulation by Hdac1, we analyzed the localized expression patterns of genes at early gastrula (*Blitz et al., 2017*). Expression profiles of Class 1 genes revealed that genes normally expressed in the vegetal region (endoderm) are significantly up-regulated in AC upon HDAC inhibition (*Figure 5F*, second bar). Coincidently, VG CRMs are hyperacetylated in AC upon HDAC inhibition (VG CRMs in *Figure 4G*), which in turn leads to the misactivation of endodermal genes in AC (*Figure 5—figure supplement 1E*, *Supplementary file 2*). Surprisingly, we found that genes normally expressed in ectoderm and mesoderm are significantly down-regulated in AC upon HDAC inhibition (*Figure 5F*, third bar) and genes normally expressed in mesoderm and endoderm are significantly down-regulated in VG (*Figure 5F*, fifth bar), upon HDAC inhibition. Contrary to the repressive role of Hdac1, this finding suggests that Hdac1 positively influences the transcription of active genes in each germ layer. We propose that the state of histone acetylation on CRMs associated with active genes is dynamic, and disruption to such equilibrium impairs normal transcription activities. Based on the quantitative analysis of pan-H3Kac profiles upon TSA treatment, we found that Hdac1-bound CRMs gain increased levels of pan-H3Kac globally including already acetylated CRMs (*Figure 4E*, *Figure 4—figure supplement 3A*). This TSA-induced excessive histone acetylation on active CRMs may contribute to the attenuated expression of associated active genes in respective germ layers (*Figure 5—figure supplement 1E*). A similar trend is observed among TSA responsive Class 3 genes (*Figure 5—figure supplement 1F*). These data demonstrate that Hdac1 not only prevents aberrant activation of silent genes, but also maintains proper gene expression levels in each germ layer (*Figure 5—figure supplement 1I,J*).

Since genes in Class 1 (CRMs having a mixture of H3K27ac and H3K27me3) and Class 3 (CRMs marked by only H3K27ac) and show localized transcriptomic profiles, we examined the expression differences between these two gene classes. Class 1 genes display a higher variability in expression levels between different germ layers when compared to Class 3 genes (*Figure 5—figure supplement 1G*). This indicates that the expression of genes undergoing active H3K27me3 suppression is more intimately associated with germ-layer determination. Altogether, these results show that Hdac1 maintains the integrity of germ-layer genes both in time and space during gastrulation.

## Discussion

Here, we defined a critical role for Hdac1 during early *Xenopus* embryogenesis. Our findings demonstrate a close link between homeostasis of histone acetylation and transcriptional activities in developing embryos. Progressive binding of Hdac1 to the genome shapes the zygotic histone acetylome, thereby reinforcing a proper germ-layer-specific transcriptome, both in time and space. Thus, Hdac1 is an essential epigenetic regulator in the control of embryonic cell identity and lineage. We propose that TFs inducing differentiation programs exploit the activity of HDACs to confine the expression of zygotic genes.

## Gradual binding of Hdac1 coincides with ZGA

The genome-wide binding of Hdac1 begins at mid-blastula and gradually accumulates at thousands of loci. Such progressive binding of Hdac1 around ZGA raises the question of whether Hdac1 recruitment requires zygotic factors during these stages. Surprisingly, ChIP-seq analyses of Hdac1 from α-amanitin-injected embryos show that zygotic transcription is dispensable for Hdac1 recruitment to the genome, at the very least, up to late blastula (*Figure 2A*, *Figure 2—figure supplement 1B, C*). Recent work in yeast shows that active transcription is required to shape histone acetylation patterns largely due to a direct role of RNAPII in recruitment and activation of H4 HATs but not HDACs (*Martin et al., 2021*). Our result agrees with the notion that Hdac1 recruitment is not directed by on-going transcription. We, therefore, speculate that maternal factors instruct early Hdac1 recruitment. Our results provide evidence that maternal Foxh1/Sox3 plays a role in Hdac1 genomic recruitment (*Figure 2E, F*), highlighting the importance of these two maternal TFs in early embryonic epigenome establishment. Though maternal TFs such as Foxh1, Vegt, and Otx1 are shown to bind the genome as early as the 32- to 64-cell stage (*Charney et al., 2017*; *Paraiso et al., 2019*), the genome-wide binding of Hdac1 likely begins at blastula and is not significantly widespread until early gastrula (*Figure 1C, G*). It is also known that Hdac1 containing complexes bind to pre-existing epigenetic marks such as DNA methylation and H3K4me3 (*Wade et al., 1999*; *Lee et al., 2018*). Therefore, maternally instructed histone modifications may also play a role in Hdac1 recruitment.

## Hdac1 functions differently on active versus inactive CRMs

One perplexing finding is that Hdac1 occupies both active and repressive genomic loci (*Figure 3A, B*). Hdac1 binds to repressive genomic regions that are facultative but not constitutive heterochromatin. Though histone modifications underlying constitutive heterochromatin have been shown to regulate developmental genes (*Riddle et al., 2011*; *Wang et al., 2018*; *Methot et al., 2021*), the profile of H3K27me3 is largely different (~90% non-overlapping peaks) from profiles of H3K9me2, 3, and H4K20me3 in early *Xenopus* development (data not shown, *van Kruijsbergen et al., 2017*). These observations suggest that Hdac1-mediated suppression is largely dictated by developmental programs. In contrast to commonly accepted repressor function of Hdac1, the binding of Hdac1 at active genomic regions is surprising. However, our findings are consistent with previous studies in yeast and mammalian cell culture emphasizing a dynamic equilibrium of histone acetylation at active loci (*Kurdistani et al., 2002*; *Wang et al., 2002*; *Wang et al., 2009*; *Kidder and Palmer, 2012*). Hence, we hypothesize that Hdac1 functions at both active and inactive CRMs in early embryos.

To test the in vivo function of HDACs, we blocked their endogenous activity using an inhibitor and quantitatively examined the changes of general H3 acetylation (pan-H3Kac) upon HDAC inhibition. First, an increase of pan-H3Kac (detecting acetylated forms of H3K9, K14, K18, and K27) is observed across Hdac1-bound CRMs, including active CRMs, consistent with the canonical enzymatic activity of Hdac1 (*Figure 4E*, *Figure 4—figure supplement 3A*). Second, repressive CRMs marked by H3K27me3 undergo drastic H3 hyperacetylation compared to active CRMs marked by H3K27ac suggesting a HDAC-activity-dependent suppression of repressive CRMs (*Figure 4F*, *Figure 4—figure supplement 3D*). Third, germ-layer-specific pan-H3Kac profiles are disrupted indicating the importance of Hdac1 in defining spatial patterns of histone acetylation (*Figure 4G*, *Figure 4—figure supplement 3I*). Based on these results, we propose a dual function model for Hdac1. On the one hand, Hdac1 prevents histone acetylation at inactive CRMs, thereby preserving H3K27 as unacetylated (*Figure 5G*). Interestingly, H3K27me3 is not always imposed on inactive CRMs. For instance, active CRMs (Cluster III, only H3K27ac) are spatially modified with pan-H3Kac (*Figure 4D*) but are not subjected to H3K27me3 (*Figure 4D*, *Figure 4—figure supplement 3I*). This suggests that HDAC-mediated histone deacetylation and Polycomb-mediated histone methylation are not coupled at inactive CRMs. On the other hand, Hdac1 participates in dynamic histone acetylation–deacetylation cycles at active CRMs (*Figure 5H*). Although we did not directly test the co-binding of HATs and HDACs, CRMs may be simultaneously bound since (1) the binding profiles of Ep300 and Hdac1 mostly overlap (*Figure 3—figure supplement 1A*), and (2) pan-H3Kac signals increase at all Hdac1 peaks including active CRMs, upon HDAC inhibition (*Figure 4E*, *Figure 4—figure supplement 3A*). Presumably, active CRMs are maintained in a state of dynamic equilibrium. This model is in accordance with a previous study demonstrating that HATs and HDACs simultaneously participate in histone acetylation cycles, which initiate and reset chromatin between rounds of transcription (*Wang et al., 2009*).

## Hdac1 safeguards misactivation of genes both in time and space

We attempted to correlate the activity of CRMs with the transcriptional activity of potential target genes. Genes associated with repressive CRMs (H3K27me3 only) are mainly inactive until neurula (*Figure 5C*). More than 85% of these genes are prematurely activated upon HDAC inhibition (*Figure 5D*), suggesting that Hdac1 maintains the state of histone hypoacetylation on repressive CRMs, thereby preventing premature expression of genes (*Figure 5—figure supplement 1H*). Moreover, genes associated with heterogeneous (both H3K27ac and H3K27me3) and active (H3K27ac only) CRMs are misactivated in different germ layers when HDAC activity is blocked (*Figure 5F*, *Figure 5—figure supplement 1F*). This indicates that Hdac1 safeguards differential histone acetylation states in each germ layer (*Figure 4G*, *Figure 4—figure supplement 3I*), restricting proper spatial transcription (*Figure 5—figure supplement 1I, J*). We did not directly address whether hypoacetylated heterogeneous CRMs are subjected to H3K27me3. However, a previous study showed that H3K27me3 is spatially deposited at late gastrula (*Akkers et al., 2009*). We predict that heterogenous CRMs are differentially marked by opposing H3K27me3 or acetylation in different germ layers. In summary, Hdac1 preserves the histone hypoacetylation state of inactive CRMs resulting in gene suppression both in time and space, thus supporting the transcriptional corepressor role for Hdac1.

## Cyclical histone acetylation sustains germ-layer gene transcription

Our study reveals an unexpected role for Hdac1 in sustaining active gene expression during germ-layer formation. Within both ectoderm and endoderm, the expression of HDAC inhibitor down-regulated genes associated with either active (H3K27ac only) or heterogeneous (H3K27ac and H3K27me3) CRMs are enriched in their respective germ layers (*Figure 5F*, *Figure 5—figure supplement 1F*). This suggests a paradoxical activator role for Hdac1, which is also reported in previous studies (*Vidal and Gaber, 1991*; *Zupkovitz et al., 2006*; *Baltus et al., 2009*; *Hughes et al., 2014*; *Rao and LaBonne, 2018*). We speculate that utilization of HDAC activities at active genomic loci is a general mechanism, as seen in examples of *Xenopus* ectoderm and endoderm lineages, which deploy distinct gene regulatory networks. Based on our findings, we propose that a dynamic equilibrium between acetylation and deacetylation is essential to sustain gene transcription. The function of HDACs on active genomic regions has been elucidated in several contexts. In yeast, cotranscriptional methylation (H3K36me3 and H3K4me2) recruits HDAC containing complexes (Rpd3S and Set3C) to suppress intragenic transcription and delay induction of genes that overlap non-coding RNAs (*Carrozza et al., 2005*; *Keogh et al., 2005*; *Li et al., 2007*; *Kim and Buratowski, 2009*; *Kim et al., 2012*; *Heo et al., 2021*). Genetic deletion of Set3C affects transcript levels only in altered growth conditions (*Lenstra et al., 2011*), consistent with the notion that cyclical histone acetylation acts as a mechanism to regulate dynamics and fidelity of transcription. In metazoans HDAC1 can be targeted by Ep300 to transcribing genes through a direct interaction (*Simone et al., 2004*). Simultaneous binding of both HATs and HDACs at active genomic regions is shown in T cells (*Wang et al., 2009*). Inhibition of both DNA methyltransferases and HDACs induces cryptic transcription in lung cancer cells (*Brocks et al., 2017*). Down-regulated genes upon HDAC inhibition exhibit high levels of cryptic transcripts during mouse cardiogenesis (*Milstone et al., 2020*). These findings suggest a role for HDAC activity in transcriptional fidelity.

Why does excessive histone acetylation due to HDAC inhibition lead to reduced transcription instead of elevated transcription? We observed that active (H3K27ac only) and heterogeneous (both H3K27ac and H3K27me3) CRMs are excessively acetylated (*Figure 4E*, *Figure 4—figure supplement 3A*) following HDAC inhibition, which results in the reduced expression of these CRM-associated genes within their respective germ layers (*Figure 5F*, *Figure 5—figure supplement 1F*). This leads us to speculate that excessive histone acetylation interferes the activity of the transcriptional machinery. A recent study shows that excessive histone acetylation on chromatin induced by inhibition of HDAC1, 2, and 3 leads to increased aberrant contacts and reduced native contacts between super-enhancer loops (*Gryder et al., 2019*). This suggests that the excessive histone acetylation impairs active transcription by altering chromatin interactions. Alternatively, excessive histone acetylation can alter the binding of acetyl-histone readers. H4 polyacetylation induced by HDAC inhibition is shown to be preferentially bound by BRD proteins (such as BRD4), thereby sequestering these factors away from active genes (*Slaughter et al., 2021*). Therefore, HDACs safeguard the function of normal acetyl-histone

readers. Further investigation of cyclical histone acetylation regulating developmental programs is needed in the context of germ-layer specification.

# Materials and methods

## Animal model and subject details

*Xenopus tropicalis* embryos were obtained by in vitro fertilization according to *Ogino et al., 2006* and staged according to *Nieuwkoop and Faber, 1994*. All embryos were cultured in 1/9× Marc's modified Ringers (MMR) at 25°C. For HDAC inhibition, 4-cell stage embryos were immersed in 1/9× MMR containing (1) 100 nM TSA (*Esmaeili et al., 2020*) or DMSO; or (2) 10 mM VPA (*Rao and LaBonne, 2018*) or $H_2O$. For α-amanitin injection, each 1-cell stage embryo was injected with 6 pg of α-amanitin (*Hontelez et al., 2015*). For morpholino injection, 10 ng of morpholino (*Foxh1* MO: 5'-TCATCCTG AGGCTCCGCCCTCTCTA-3', *Chiu et al., 2014*; *Sox3* MO: 5'-GTCTGTGTCCAACATGCTATACATC-3', *Gentsch et al., 2019*) was injected into 1-cell staged embryos. For spatial analyses, embryos were dissected at the late blastula stage (6 hpf), and explants were cultured to the early gastrula (7 hpf). Animals were raised and maintained following the University of California, Irvine Institutional Animal Care Use Committee (IACUC). Animals used were raised in the laboratory and/or purchased from the National *Xenopus* Resource (RRID: SCR_013731).

## Western blotting

Embryos were homogenized in 1× RIPA (50 mM Tris–HCl pH7.6, 1% NP40, 0.25% Na-deoxy-cholate, 150 mM NaCl, 1 mM etheylenediaminetetraacetic acid [EDTA]), 0.1% sodium dodecyl sulfate [SDS], 0.5 mM dithiothreitol (DTT) with protease inhibitors (Roche cOmplete) and centrifuged twice at 14,000 rpm. The supernatant was then subjected to western blotting using anti-HDAC1 (Cell Signaling, 34589S), anti-HDAC2 (Genetex, GTX109642), and anti-Tubulin (Sigma, T5168). For histone modifications, acid-extracted histone lysate was prepared accordingly (*Shechter et al., 2007*) and subjected to western blotting using anti-H3K9ac (Cell Signaling, 9649), H3K14ac (Cell Signaling, 7627), H3K18ac (Cell Signaling, 13998), H3K27ac (Cell Signaling, 8173), H3K56ac (Cell Signaling, 4243), and H4K20ac (Active Motif, 61531).

## ChIP and ChIP-seq analysis

ChIP protocol was performed as described (*Chiu et al., 2014*). Antibodies used for ChIP were anti-HDAC1 (Cell Signaling, 34589S, 1:100), anti-HDAC2 (Cell Signaling, 57156S, 1:100), anti-H3K18ac (Cell Signaling, 13998, 1:100), and anti-Sox3 (*Zhang et al., 2003*). ChIP-seq libraries were constructed using the NEBNext Ultra II DNA Kit (NEB, E7645).

For sequential ChIP, the first round of ChIP was performed as described and eluted in 1× Tris-EDTA (TE containing 1% SDS at 37°C for 30 min). The eluate was diluted ten-fold with 1× RIPA (without SDS) and subjected to the second round of ChIP as described (*Desvoyes et al., 2018*). Real-time quantitative PCR (RT-qPCR) was performed using Power SYBR Green PCR master mix (Roche) to quantify the DNA recovery compared to ChIP input DNA at one embryo equivalency (percent input). The error among technical replicates was calculated using the rule of error propagation. ChIP qPCR primer sequence information is provided in *Supplementary file 3*.

For dissected pan-H3Kac ChIP, spike-in chromatin (Active Motif, 53083) was added to the chromatin of dissected tissues at a ratio of 1:35. Mixed chromatin was then subjected to ChIP with 5 µg anti-panH3Kac (Active Motif, 39139) and 1 µg of anti-H2Av (Active Motif, 61686) and followed as described.

All experiments were performed in two independent biological replicates unless noted. Sequencing was performed using the Illumina NovaSeq 6000 and 100 bp single-end reads or 100 bp paired-end reads were obtained.

All sequencing data were aligned to *Xenopus tropicalis* v10.0 genome (http://www.xenbase.org/, RRID:SCR_003280) using Bowtie2 v2.4.4 (*Langmead and Salzberg, 2012*). PCR duplicates were removed using Samtools v1.10 (*Li et al., 2009*). ChIP-seq signals were visualized using IGV v2.11.3 (*Robinson et al., 2011*) after concatenating two biological replicates when available. IDR analysis (*Li et al., 2011*) was used to identify high-confidence peaks called by Macs2 v2.7.1 (*Zhang et al., 2008*) against the stage-matched input (*Charney et al., 2017*) between two biological replicates according

to ENCODE3 ChIP-seq pipelines (IDR threshold of 0.05) (https://docs.google.com/document/d/1lG_Rd7fnYgRpSIqrIfuVlAz2dW1VaSQThzk836Db99c/edit). Differential ChIP peak analysis was performed using Homer v4.11 (*Heinz et al., 2010*).

For dissected pan-H3Kac ChIP, all second replicates were downsampled to 25% to compare equivalent sequencing depth. *Drosophila* H2Av peaks are generated from published S2 cell samples (*Tettey et al., 2019*). Normalization factors were then calculated based on reads that mapped to *Drosophila* H2Av peaks for each ChIP-seq sample (*Egan et al., 2016*). Detailed normalization factors used are listed *Supplementary file 4*.

## RNA-seq and analysis

Total RNA from dissected tissues was extracted using Trizol as described (*Amin et al., 2014*). mRNA was then isolated using NEBNext PolyA mRNA Magnetic Isolation Module (NEB E7490S). Sequencing libraries were prepared using NEBNext Ultra II RNA library prep kit (NEB E7770S) and sequenced by the Illumina NovaSeq 6000 with 100 bp paired-end reads. All experiments were performed in two independent biological replicates.

All sequencing samples were aligned using STAR v2.7.3a (*Dobin et al., 2013*) to *Xenopus tropicalis* genome v10.0 (http://www.xenbase.org/, RRID:SCR_003280) to obtain raw read counts. RSEM v1.3.3 (*Li and Dewey, 2011*) was used to calculate expression values in transcripts per million (TPM) which are used to construct heatmaps depicting gene expression levels. Differentially expressed genes were identified using edgeR v3.36.0 (*Robinson et al., 2010*) with the following parameters: greater than twofold change and less than 0.05 false discovery rate (also known as the adjusted p-value), in R v4.1.2 (*R Development Core Team, 2021*). Metascape (*Zhou et al., 2019*) was used to perform gene ontology enrichment analyses with default parameters (min overlap = 3, p-value cutoff = 0.01, and min enrichment = 1.5).

## Additional bioinformatics and statistical analyses

Samtools v1.10 (*Li et al., 2009*) was used to convert between SAM and BAM files. DeepTools v3.5.0 (*Ramirez et al., 2014*) was used to generate: (1) ChIP-seq signal track (bigwig files) normalized by reads per genomic content (-RPGC) at the bin size of 1 bp; (2) heatmaps around peak summits normalized by Bins Per Million mapped reads (-BPM) at the bin size of 50 bps; (3) signal profile along the gene bodies normalized by -BPM at the bin size of 50 bps; (4) Pearson correlation between ChIP-seq samples at peaks. Homer v4.10 (*Heinz et al., 2010*) was used to annotate genomic features of ChIP peaks. Bedtools v2.29.2 (*Quinlan and Hall, 2010*) was used to determine peak overlaps among ChIP-seq peaks and obtain counts of reads at ChIP-seq peaks. CentriMo (*Bailey and Machanick, 2012*) was used to perform local motif enrichment analysis. Welch Two-sample *t*-test in R v4.1.2 was used to determine the statistical significance between groups. Cohen's *d* (effect size) was calculated using lsr v0.5.2 package in R v4.1.2 (*R Development Core Team, 2021*). Time-course gene expression was obtained from ribosomal RNA-depleted RNA-seq data (*Owens et al., 2016*). The expression in TPM was calculated as outlined above. Spatial gene expression at early gastrula was obtained from RNA-seq of five dissected tissues (*Blitz et al., 2017*) consisting of the animal cap (ectoderm), the dorsal marginal zone (dorsal mesoderm), the lateral marginal zone (lateral mesoderm), the ventral marginal zone (ventral mesoderm), and the vegetal mass (endoderm). The expression in TPM was obtained as outlined above. Fisher's exact test (alternative = "greater") in R v4.1.2 (*R Development Core Team, 2021*) was used to determine the significance of the proportional enrichment between groups. p-Values from Fisher's exact test are summarized in *Supplementary file 5*.

## Categorical analyses

Spatial categorization of CRMs is defined as below: AC CRMs represent CRMs whose pan-H3Kac signals in AC is twofold higher than in VG; VG CRMs represent CRMs whose pan-H3Kac signals in VG is twofold higher than in AC; ubiquitous CRMs represent the remaining CRMs whose pan-H3Kac signals do not exceed twofold enrichment in either of the two examined germ layers. For temporal gene expression analysis, (strictly) zygotic genes are determined by removing (1) genes whose expression levels are greater than 1 TPM during the first 2 hr post-fertilization and (2) genes whose expression levels are less than 1 TPM from 0 to 23 hpf. Spatial categorization of genes at early gastrula stage: the average TPM between three dissected mesoderm tissues (dorsal, marginal, and lateral marginal

zones) was used to represent the expression of mesoderm. Genes with the expression in any dissected tissue less than 1 TPM were considered not expressed. Genes with the coefficient of variance of TPM less than 0.1 (10%) were considered evenly expressed. The remaining genes with localized expression were assigned to a germ layer based on the maximum TPM.

## Acknowledgements

We thank Drs. Kyoko Yokomori (University of California, Irvine), Yongsheng Shi (University of California, Irvine), and current Cho lab members for insightful comments on this study. We thank the Genomic High Throughput Facility at the University of California, Irvine for sequencing services. We also thank the Research Cyberinfrastructure Center at the University of California, Irvine for the ongoing support of High Performance Community Computing Clusters. This work is supported by NIH R01GM126048 and ACS RSG-18-009-01-CCG to WW and NIH R01GM126395, R35GM139617 and NSF 1755214 to KWYC.

## Additional information

### Funding

| Funder | Grant reference number | Author |
|---|---|---|
| National Institute of General Medical Sciences | R01GM126395 | Ken WY Cho |
| National Institute of General Medical Sciences | R35GM139617 | Ken WY Cho |
| National Science Foundation | 1755214 | Ken WY Cho |
| National Institute of General Medical Sciences | R01GM126048 | Wenqi Wang |
| American Cancer Society | RSG-18-009-01-CCG | Wenqi Wang |

The funders had no role in study design, data collection, and interpretation, or the decision to submit the work for publication.

### Author contributions

Jeff Jiajing Zhou, Conceptualization, Data curation, Formal analysis, Validation, Investigation, Visualization, Methodology, Writing - original draft; Jin Sun Cho, Han Han, Investigation; Ira L Blitz, Resources, Writing - original draft, Writing - review and editing; Wenqi Wang, Resources, Investigation; Ken WY Cho, Conceptualization, Supervision, Funding acquisition, Investigation, Writing - original draft, Writing - review and editing

### Author ORCIDs

Wenqi Wang (ID) http://orcid.org/0000-0003-4053-5088
Ken WY Cho (ID) http://orcid.org/0000-0001-7282-1770

### Ethics

All of the animals were handled according to approved Institutional Animal Care and Use Committee (IACUC) protocols (#AUP-21-068) of the University of California, Irvine.

### Decision letter and Author response

Decision letter https://doi.org/10.7554/eLife.79380.sa1
Author response https://doi.org/10.7554/eLife.79380.sa2

## Additional files

### Supplementary files

• Supplementary file 1. Top transcription factor binding motifs detected in HDAC1 peaks at stage 8, 9 and 10 embryos.

• Supplementary file 2. References for known germ-layer functioning genes in *Figure 5—figure supplement 1E*.

• Supplementary file 3. ChIP-qPCR Primer Sequences.

• Supplementary file 4. ChIP spike-in normalization.

• Supplementary file 5. p-Values from Fisher's exact test.

### Data availability

Raw and processed RNA-seq and ChIP-seq datasets generated from this study are available at NCBI Gene Expression Omnibus using the accession GSE198378. Publicly available datasets used in this study are available at NCBI Gene Expression Omnibus using the accession GSE56000 (*Gupta et al., 2014*; H3K27ac ChIP-seq), GSE67974 (*Hontelez et al., 2015*; Ep300, H3K9ac, H3K4me1, H3K4me3, H3K36me3, H3K9me2, H3K9me3, H3K27me3, H4K20me3 ChIP-seq, and st12 RNA Pol2 ChIP-seq), GSE65785 (*Owens et al., 2016*; temporal profiling of RNA-seq), GSE85273 (*Charney et al., 2017*; st9 Foxh1 ChIP-seq, st7, 8, 9, and 10.5 RNA Pol2 ChIP-seq), GSE81458 (*Blitz et al., 2017*; st10.5 dissected germ-layer RNA-seq), and GSE129236 (*Tettey et al., 2019*; H2Av ChIP-seq in S2 cells). Relevant bioinformatic analysis scripts are accessible at https://github.com/jiajinglz/bioRxiv_05052022_Hdac_dual_roles (copy archived at *Zhou, 2023*).

The following dataset was generated:

| Author(s) | Year | Dataset title | Dataset URL | Database and Identifier |
|---|---|---|---|---|
| Zhou JJ | 2022 | Histone deacetylase 1 maintains lineage integrity through histone acetylome refinement during early embryogenesis | https://www.ncbi.nlm.nih.gov/geo/query/acc.cgi?&acc=GSE198378 | NCBI Gene Expression Omnibus, GSE198378 |

The following previously published datasets were used:

| Author(s) | Year | Dataset title | Dataset URL | Database and Identifier |
|---|---|---|---|---|
| Gupta R, Baker JC | 2014 | Enhancer chromatin signatures predict Smad2/3 binding in *Xenopus* | https://www.ncbi.nlm.nih.gov/geo/query/acc.cgi?acc=GSE56000 | NCBI Gene Expression Omnibus, GSE56000 |
| Hontelez S, Veenstra GC | 2015 | Embryonic transcription is controlled by maternally defined chromatin state | https://www.ncbi.nlm.nih.gov/geo/query/acc.cgi?acc=GSE67974 | NCBI Gene Expression Omnibus, GSE67974 |
| Owens ND, Blitz IL, Lane MA, Patrushev I, Overton JD, Gilchrist MJ, Cho KW, Khokha MK | 2016 | Measuring Absolute RNA Copy Numbers at High Temporal Resolution Reveals Transcriptome Kinetics in Development | https://www.ncbi.nlm.nih.gov/geo/query/acc.cgi?acc=GSE65785 | NCBI Gene Expression Omnibus, GSE65785 |
| Charney RM, Cho KW | 2017 | Foxh1 marks the embryonic genome prior to the activation of the mesendoderm gene regulatory program | https://www.ncbi.nlm.nih.gov/geo/query/acc.cgi?acc=GSE85273 | NCBI Gene Expression Omnibus, GSE85273 |
| Blitz IL, Paraiso KD | 2017 | Regional expression of *X. tropicalis* transcription factors in early gastrula embryos | https://www.ncbi.nlm.nih.gov/geo/query/acc.cgi?acc=GSE81458 | NCBI Gene Expression Omnibus, GSE81458 |

*Continued on next page*

*Continued*

| Author(s) | Year | Dataset title | Dataset URL | Database and Identifier |
|---|---|---|---|---|
| Tettey TT, Gao X, Shao W, Li H, Story BA, Chitsazan AD, Glaser RL, Goode ZH, Seidel CW, Conaway RC, Zeitlinger J, Blanchette M, Conaway JW | 2019 | Expression profiling by high throughput sequencing Genome binding/ occupancy profiling by high throughput sequencing | https://www.ncbi. nlm.nih.gov/geo/ query/acc.cgi?acc= GSE129236 | NCBI Gene Expression Omnibus, GSE129236 |

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
