## [Editor Report]

In this revised and important study, the authors investigate the roles of histone deacetylases in the spatial epigenetic regulation of zygotic gene expression during embryo gastrulation. They provide convincing evidence for HDAC1 binding to genes around the timing of large-scale genome activation, and that inhibition of histone acetylation blocks gastrulation, blurring cell lineage integrity, tied to both positive and negative regulatory effects on transcription in space and time. The research reveals new insight on the role of histone acetylation-deacetylation in dynamics in epigenetic control of gene expression and cell fate determination during early tissue patterning in embryogenesis.

---

## [Decision Letter]

**Decision letter after peer review:**

Thank you for submitting your article "Histone deacetylase 1 maintains lineage integrity through histone acetylome refinement during early embryogenesis" for consideration by *eLife*. Your article has been reviewed by 3 peer reviewers, one of whom is a member of our Board of Reviewing Editors, and the evaluation has been overseen by Jessica Tyler as the Senior Editor. The reviewers have opted to remain anonymous.

Essential revisions:

The reviewers highlighted a number of novel features of your study. This includes uncovering dual positive and negative roles for HDACs in cell fate specification in the early embryo and identifying a key contribution of HDACs to germ layer cell lineage integrity. Inhibition of deacetylation causes dysregulation of specificity of spatial boundaries in the gastrula, with mesendoderm misexpressed in animal cap cells. Additionally, you demonstrate progressive binding of HDAC1 to the genome with ChIP-seq and find correlative patterns of promoter occupancy relative to other repressive and activating histone epigenetic marks and ChIP-seq peaks for maternal transcription factors, FoxH1 and Sox3. On the whole, this work advances our understanding of HDACs and control of chromatin acetylation and gene expression tied to early cell fate specificity in a vertebrate embryo.

However, the manuscript is not acceptable in its current format and will require major experimental revisions for consideration. Referees brought up a number of concerns.

1) Direct versus indirect effects of deacetylation inhibition. Specifically, whether H3-acetylation peaks and gene expression changes from deacetylase inhibition can be directly assigned to HDAC1 activity. One reviewer suggested a knockdown of HDAC1 would be useful, as performed in a previous study by Rao and Labonne, Development 2018. Additionally, because the authors primarily performed their experiments with TSA, rather than HDAC1-specific VPA, it is suggested that they expand the comparison, at least for RNA-seq, and also consider directly comparing the effects on histone to separately assign chromatin binding and gene regulation individually to HDAC1 versus HDAC2.

2) Lack of evidence that HDAC1 recruitment is likely dependent on maternal factors. The authors relied on the correlation of HDAC1 ChIP-seq peaks from this manuscript to FoxH1 and Sox3 peaks in published ChIP-seq datasets. Although the correlation is supportive, reviewers felt that the conclusion lacked experimental support. They proffer two suggestions: (1) to temporally analyze when HDAC1 binding begins in the egg or early gastrula, or (2) to knockdown maternal FoxH1 and Sox3 and characterize the loss of HDAC1 peaks from ChIP-seq.

3) There are a number of comments from reviewer 2 about improving manuscript clarity, and specific notes from reviewer 3 about analyses and figures. These should be addressed as well. Reviewer 1 suggested highlighting further that TSA causes more extensive gene expression dysregulation in AC versus VG, and to specifically plot mis-expression of endoderm VG genes in AC.

*Reviewer #1 (Recommendations for the authors):*

This study provides a number of new insights on the role of histone acetylation-deacetylation in spatial regulation of gene expression that contributes to cell lineage regulation as germ layers begin to form in the gastrula embryos. The work is important and interesting to the field. However, my enthusiasm is partly limited by the correlative nature of the experiments and some missing analyses. My comments to the author are below, formatted as ways to improve the manuscript.

1. Extent of mis-regulation of germ layer-specific genes upon TSA and VPA expression. The authors provide volcano plots in Figure S5B, and differential gene expression analysis spatially within AC and VG in Figure 4F. They find that up-regulation of H3Kac peaks tends to occur for opposite lineage, ie. VG genes are misexpressed in AC. Further, that downregulated genes tend to be more lineage constrained. It would be helpful if they could also plot a smaller subset of the data for ectoderm and endoderm-specific gene lists. Based on Figure 4F, one would expect that a large fraction of endoderm-specific genes would be inappropriately upregulated in the AC, and a large fraction of ectoderm-specific genes upregulated in the VG. To what extent is this true.

I also think the authors should highlight and discuss that TSA treatment has a much more profound effect on gene expression dysregulation in the AC compared to the VG (Figure S5B). There are many more genes up and downregulated in the AC. This is quite a novel and interesting result. Can the authors comment on why?

2. Timing of HDAC1 binding to the genome. It is interesting that HDAC1 peaks are widespread in the genome by Stage 8, presumably during zygotic genome activation, and that these peaks largely strengthen over time into the early gastrula. This however begs the question of when HDAC1 begins to bind to the genome. Is it already bound in the egg, or upon fertilization, or are peaks only detectable starting around the time of genome activation? Given that the protein is present in the egg and throughout cleavage stages (Figure 1A), and that the authors propose FoxH1 and Sox3 may be contributing to recruitment, the manuscript would be strengthened by characterization of the time of recruitment.

3. Differential role for HDAC1 and HDAC2 in lineage maintenance? The TSA treatment makes it difficult to disentangle specific subsets of genes modulated by promoter-proximal deacetylation. Can the authors contrast TSA vs. VPA inhibition experiments to identify peaks that could likely be attributed to HDAC2 rather than HDAC1, and perform GO analyses? Alternatively, can the authors provide direct binding data with ChIP-seq of HDAC2 in blastula and gastrula.

4. Direct evidence that HDAC1/2 binding contributes to gene expression regulation. The authors utilize inhibition of deacetylation (TSA) or HDAC1 specifically (VPA) to analyze changes to pan-H3K-acetylation and gene expression, regionally within the AC and VG. In general, I like this line of experiments, however, I wonder whether the TSA inhibition might be affecting other deacetylase enzymes, and what the effect would be of MO knockdown of the HDACs (or their recruitment via FoxH1, Sox3). Other groups have found that HAT-HDAC binding to promoters and enhancers does not always correlate to gene expression regulation. Extending this notion here, I worry that the authors may not be directly analyzing the downstream function of HDACs. Can the authors compare their differential expression data to existing gene profiling studies in embryos that have knockdown of FoxH1, Sox3, or HDAC1/2 to demonstrate similar profiles of dysregulation?

*Reviewer #2 (Recommendations for the authors):*

As a general reader, I do not know enough of the background knowledge in the field to know how novel the observations are, and it seems that most of these specific comparisons are made in the discussion, and it is not explicit what observations are really new, or whether the importance of the paper is in the documentation. The authors need to draw this out more clearly. Also, as a general reader without the ingrained knowledge of the activating or inhibitory functions of the different histone modifications, I initially found it impossible to hold onto the threads of the arguments but was able to do so by color coding the modification types as activating, inhibitory, or heterochromatin in the text. I would hope that modern journals might be able to use colored text to enable readers to read and evaluate the text more readily. Similarly, it is hard to hold on to the nature of different classes of response, so I suggest more descriptive names for the classes. Without these, the manuscript is very tough going for the general reader. Indeed, as a detailed examination of chromatin modification, it is inevitably dense. Again as a general reader, I would find it more useful to present the specific genes that convey the arguments in the main text and figures, so that there is something to get one's teeth into and move the venn diagrams and stacked bar graphs and violin plots, which I find uninformative for the specific arguments, but which are necessary for the generality of the claims, to the supplement. But these are suggestions to make the manuscript more accessible to the general reader. If the target audience is the specialized field of chromatin organization or detailed regulation of gene expression in developing animals (without going beyond inhibitor studies to the sequences responsible for the DNA recruitment or the enzymatic activity of the protein), then the paper would probably be more appropriate for a more focused journal. But if there are new insights that can be more clearly articulated, then the manuscript could be appropriate to *eLife*.

Major points, in no particular order.

1. While the Hdac genes are shown for mRNA it would be useful to see whether this matches the protein databases. Are they detectable, and does mRNA translate to protein abundance?

2. Line 140- were Vegt peaks found to be enriched in the maternal motifs? (Gentsch?)

Indeed, Was there enrichment of other motifs associated with transcriptional activation as found by Gentsch et al? (POU-SOX (Pou5f3-Sox3 heterodimer), Krüppel-like zinc finger (ZF; Sp1 and several Klf), POU (Pou5f3), SOX (Sox3), bZIP (Max), FOXH (Foxh1), ETS (Ets2), NFY (NFYa/b/c), SMAD (Smad1/2), VegT (mVegT)), A lack of enrichment would illustrate a strong difference in recruitment and be informative.

3. "Consistent with the overlapping binding of Ep300, Hdac1 peaks display a moderate level of H3K9ac (Cluster c and d), high levels of 164 H3K18ac (Cluster b, c, and d), H3K27ac (Cluster b, c, and d), and pan-H3 lysine acetylation (pan-H3Kac)."

Why is this consistent (ref required?) is Ep300 always an activating signal? The next section discusses activating signals, so maybe so, but it does not say so. What is the current understanding of Ep300? General enhancer binding? I fear that the paper is exceedingly hard to follow unless one has these identities imprinted in one's own brain, which I don't. Indeed, for people not in the immediate field, the nomenclature, with its acetylations, and multiple methylations is all too easy to forget. For ease of reviewing, I color-coded the histone marks according to Wikipedia. It made the manuscript much easier to follow, though I am sure my simplistic green or red color code is too simple. The color coding of the figures in the manuscript is more nuanced, and presumably reflects some deeper understanding. It would be useful to share this with the reader. Otherwise, the activating (green) repressive (red), and heterochromatin (brown, blue?- what are these?) and the main text should also match an informative color coding, which should be trivial in these days of digital typesetting.

4. "we speculate that these regions are differentially marked in space due to heterogeneous cell populations present in the whole embryo. Hdac1 Cluster I peaks are referred to as heterogeneous CRMs."

Surely the authors can give explicit examples here for some known heterogeneously expressed genes to reduce the speculative nature of this group? (though they do eventually show up in the supplements- but why not progress the argument from the specific to the general?).

5. Ideally, the figures and legends should be interpretable independent of the text, for example, when wondering what abcd are in figures 1 and 3, one should not have to search in the text for the relevant explanation.

"Overall, a minority of Hdac1 peaks are unique to each of the blastula stages (Cluster a and b) while a majority of Hdac1 peaks are present across multiple stages (Cluster c) and at the early gastrula stage (Cluster d) (Figure 1C, S1D).”

And in reading this, "unique to each blastula stage", could be expressed more clearly to indicate their uniqueness- ie at stage 8 (cluster a, likely driven by maternal factors?) and stage 9, (cluster b, likely driven by newly expressed zygotic factors or signals?).

Re the italics – where are the genes known to be in these classes? Ie the Nodals, vs genes driven by zygotically activated signals like Smad2? It is important for a simple argument to give the general reader some specific biological context.

6. Likewise "These observations suggest that Hdac1-mediated suppression is largely dictated by developmental programs" – perhaps the authors might give some examples in the text.

7. "First, an increase of pan-H3Kac is observed across Hdac1-bound CRMs, including active CRMs, consistent with the canonical enzymatic activity of Hdac1" remind us what H3 Kac is for – it seems to be in the purple, brown and red categories. Is it associated with transcription start sites? It's not clear what antibody was used. It seems it must be a pan H3Kac antibody?

8. What are the effects of Hdac inhibitors on the transcriptome? Seems that this is an essential piece of information to interpret the results on chromatin? Seems there is a lot of literature on this, and so far as I can tell it is not cited until the discussion. It would be nice to have a digested understanding to compare to the chromatin results in the Results section (some of this appears eventually, but late enough to leave the reader frustrated when the topic comes up).

9. I had great difficulty understanding the paragraph that begins "Given that Hdac1 CRM clusters (Clusters I-IV in Figure 3C) are subjected to both active and repressive epigenetic modifications presumably in different germ layers, we compared the general status of H3 acetylome (pan-H3Kac) between two distinct germ layers" I think that all of this is VPA and TSA independent information? But in the end, isn't it all circular? Ie higher enrichment of pan H3Kac associated with higher expression?

10. "Taken together, these data demonstrate that Hdac1 maintains differential H3 acetylation states between germ layers through its catalytic activity." Is a rather bland statement after trudging through the paragraph. I think that one can make the general statement that upon inhibition of HDAC, repressed genes have a larger increase in their pan H3Kac than do active genes. And is there evidence that it is through the catalytic activity rather than its binding?

"we found that the fold increases of pan-H3Kac signals after TSA treatment are negatively correlated with the levels of endogenous pan-H3Kac signals in both AC and VG explants" This is an unnecessarily complicated sentence. It's already hard to figure out the direction things are going in without having to think about what negative correlation means in these contexts. But I think it means what I just suggested?

11. I ask that the authors consider how much time would be spent by the reader looking back to see what Class1,2 and 3 genes are. Could they not be more readily described as "selectively activated", "constitutively repressed" and "Constitutively activated" if these were designated SA, CR, and CA that might be easier to remember/interpret, especially for me who can't remember class I and 2 from one paragraph to the next. Or consider the value of spelling them out each time, at least for the reviewer. Once accepted, one can return to uninformative abbreviations and let the reader suffer.

12. What is the mechanism of Hdac inhibition by VPA and TSA? Does the Hdac dissociate from chromatin, allowing access to acetylases? Or does it stay on the chromatin in an inactive form, suggesting a more complex mechanism of the differential effects of inactivation? What do the data say about the potential mechanism?

*Reviewer #3 (Recommendations for the authors):*

My general concern regarding this work is that most of the conclusions come from indirect evidences:

– Maternal TF involvement is only supported by motif enrichment.

– Spatial difference in H3K27ac/me and HDAC1 marking between AC/V part of embryos based on whole embryos data.

– Functional test of HDAC1 requirement carried out through the use of broad HDAC inhibitor. Here I find that the authors investigate in great detail the behaviour of HDAC1-associated regions but do not provide evidence that these regions are preferentially affected by TSA treatment. HDAC1 knockdown would be a much more straightforward assay to interpret in my view.

Specific points

Figure 1

1C: St101/2 or stage 9-specific HDAC1 peaks seem to already show signal at stage 8. (Difficult to grasp relative level as no indication of what the blue scale is….). In any case, is this low level at stage 8 meaningful (how does it look in all enhancer peaks for example).

What is the temporal expression (early blastula to neurula stages for example) of the a-d set of HDAC1-associated genes? Any correlation between stage-specific binding and expression pattern would strengthen the proposed involvement of HDAC1 in the epigenetic regulation of genes around ZGA.

Figure 2:

2B What are the enriched motifs in stage 8 and stage 10 specific peaks?

2C: How does the HDAC1 peak intensity look if you integrate the HDAC1 chip a-amanitin data in the map from 1C (in complement to S2B).

Charney et al. document binding of TF pre/post ZGA.

To support the ID that TF binding is associated with HDAC1 binding it would be important to check if stage-specific binding of HDAC1 is associated with stage-specific binding of TF.

The title "HDAC1 binding is instructed maternally" is not appropriate. Interference with maternal TFs would be needed to go beyond correlation. HDAC1 binding after foxh1MO (as in Charney et al) will indicate if HDAc1 binding depends on the presence of this maternal TF

Figure 3

"moderate" level of K9ac etc….moderate compared to what??? I would think comparing for example level of p300+ HDAC+ to p300+HDAc- would give an actual idea if the level on HDAC associated p300 peak stands out.

The same goes for figure S3B.

Without such kind of comparison, it is difficult to conclude that HDAC1 peaks have biased epigenetic status compared to all p300 peaks for example.

Again in Figure 3B HDAc1 bound region exhibits a strong signal density of K27me3.

Line 183: I don't think cluster b fits the description.

Fig3CD why exclude HDAC peaks from stage 8?

It would be useful to see how HDAC peaks signal looks on Figure 3D map.

In 3D there seems to be a clear anti-correlation between K27me3 and K27ac signal (see cluster I) that does not really fit the description of line187-191. This is important because the concept proposed in the manuscript relies on these co-occurring K27ac/me sites. Are the signal anti-correlated in cluster I?

S3C K27ac and me do not seem to overlap much. Can we see the region where peaks are detected (add a box where peaks are detected)?

Line 203 – before testing of this hypothesis is surely again to compare acetylation in p300+ HDAC+ versus p300+ HDAC- (or at TSS) to evaluate if TSA primarily affects HDAc1 sites?

Figure 4A/B

Can the author use a TSA control experiment where TSA is applied only prior to ZGA or only around ZGA?? This will indicate what is the time window when TSA has this effect.

It would be also important to document the effect on the development of whole embryos of the TSA treatment scheme used on explant (Figure 4C). Are the embryos defective in these conditions?

4C:

What is the effect of TSA on the number of peaks detected?

Would be very important to have a heat map that summarizes the panAc-peak in control and TSA conditions as well as the HDAc1 peak. This would illustrate to which extent TSA primarily affects HDAC binding sites.

AC and VG-specific peaks seem to exhibit a much lower intensity of panAc signal than common (scale missing on the map…). What could be the reason for such a difference?

4E FC I assume is FC TSA / control.

Fig4E and S4G

Global increase in acetylation in embryos (4D) – immediate question whether acetylation on the genome disproportionately affects HDAC1 binding sites?

This is critical since the authors assume that TSA effect indeed reflects the effect on HDAC1 activity. Authors, unfortunately, focus analysis solely on HDAc1 binding sites so the conclusion line249 is not yet supported.

All observations could also fit the model whereby TSA leads to the homogeneous increase of ac level genome-wide – would obviously lead to lower fold change if starting point high (cluster III) and higher if low (cluster II).

Figure 5

When excluding genes with maternal transcripts how many genes are left in each category?

How do sets of genes with the same epigenetic configuration but HDAC- compare (i.e genes that are K27me3/K27ac but without HDAC1?).

VPA analysis cannot support the claim that TSA treatment has no off-target effect because this (i) is not RNA-seq analysis and (ii) 8/24 genes selected for RTqPCR are not showing the expected effect.

The first important question to address: are the TSA-affected genes disproportionally enriched for HDAC peaks associated genes?

5E Not sure to get the significance of observation.

Can it be that a gene can be detected as upregulated only if they have a low level in control conditions can be detected as downregulated only if they have a high level in control conditions? Which is the rationale for the selection of Ac or V-specific genes…?

---

## [Author Response]

1) Direct versus indirect effects of deacetylation inhibition. Specifically, whether H3-acetylation peaks and gene expression changes from deacetylase inhibition can be directly assigned to HDAC1 activity. One reviewer suggested a knockdown of HDAC1 would be useful, as performed in a previous study by Rao and Labonne, Development 2018. Additionally, because the authors primarily performed their experiments with TSA, rather than HDAC1-specific VPA, it is suggested that they expand the comparison, at least for RNA-seq, and also consider directly comparing the effects on histone to separately assign chromatin binding and gene regulation individually to HDAC1 versus HDAC2.

We followed the advice and examined the effect of inhibiting Hdac activities using both TSA and VPA (Figure S4.1-3 Figure 5F, Figure S5A-F). Additionally, we also performed Hdac2 ChIP-seq to compare the chromatin binding behavior of Hdac1 and Hdac2 (Figure S1E-H).

2) Lack of evidence that HDAC1 recruitment is likely dependent on maternal factors. The authors relied on the correlation of HDAC1 ChIP-seq peaks from this manuscript to FoxH1 and Sox3 peaks in published ChIP-seq datasets. Although the correlation is supportive, reviewers felt that the conclusion lacked experimental support. They proffer two suggestions: (1) to temporally analyze when HDAC1 binding begins in the egg or early gastrula, or (2) to knockdown maternal FoxH1 and Sox3 and characterize the loss of HDAC1 peaks from ChIP-seq.

Inhibition of zygotic transcription using a-amanitin injection showed that the genomic binding profile of Hdac1 is largely unchanged at the late blastula stage. This inhibition of zygotic gene expression provides strong evidence supporting the conclusion that zygotic factors are not required for Hdac1 binding at this stage, implicating maternal recruitment. Additionally, we now include the ChIP-qPCR analysis of Hdac1 binding around Foxh1/Sox3 cobound regions in Foxh1 or Sox3 morphants. We found that Hdac1 binding is reduced, but not abolished in these morphants, suggesting that, in addition to Foxh1 and Sox3, other maternal TFs are likely to be involved in the Hdac1 recruitment to DNA.

3) There are a number of comments from reviewer 2 about improving manuscript clarity, and specific notes from reviewer 3 about analyses and figures. These should be addressed as well. Reviewer 1 suggested highlighting further that TSA causes more extensive gene expression dysregulation in AC versus VG, and to specifically plot mis-expression of endoderm VG genes in AC.

We incorporated all suggestions raised by reviewers and also included additional gene expression dysregulation in AC vs VG (Figure S5E, Table S3).

Reviewer #1 (Recommendations for the authors):This study provides a number of new insights on the role of histone acetylation-deacetylation in spatial regulation of gene expression that contributes to cell lineage regulation as germ layers begin to form in the gastrula embryos. The work is important and interesting to the field. However, my enthusiasm is partly limited by the correlative nature of the experiments and some missing analyses. My comments to the author are below, formatted as ways to improve the manuscript.1. Extent of mis-regulation of germ layer-specific genes upon TSA and VPA expression. The authors provide volcano plots in Figure S5B, and differential gene expression analysis spatially within AC and VG in Figure 4F. They find that up-regulation of H3Kac peaks tends to occur for opposite lineage, ie. VG genes are misexpressed in AC. Further, that downregulated genes tend to be more lineage constrained. It would be helpful if they could also plot a smaller subset of the data for ectoderm and endoderm-specific gene lists.

We added a heatmap showing the germ layer specific gene expression changes upon HDAC inhibition by TSA and VPA (Figure S5E). Table S3 lists functions of these genes and their PubMed references.

Based on Figure 4F, one would expect that a large fraction of endoderm-specific genes would be inappropriately upregulated in the AC, and a large fraction of ectoderm-specific genes upregulated in the VG. To what extent is this true.

Figures 5F and S5F support the notion that a large fraction of endoderm-specific genes are inappropriately upregulated in the AC, whereas a large fraction of ectoderm-specific genes are upregulated in the VG upon HDAC inhibition.

To determine whether endodermal genes are preferentially upregulated in ectodermal cells upon TSA treatment (Figure 5F), we compared the following genomic data sets: (1) a list of genes preferentially expressed in ectodermal cells upon HDAC inhibition, (2) a list of endodermal genes based on dissected gastrula RNA-seq data (Blitz et al., 2017). We used Fisher’s exact test to determine genes are upregulated in ectodermal lineage cells upon HDAC inhibition.

The parameters to assign genes to spatial expression patterns is described in Materials and methods under Categorical Analyses. Genes associated with low expression levels (<1 TPM) are categorized as not expressed (NE), genes whose coefficient of variation is less than 10% among the 3 germ layers are considered expression not localized (NL), genes expressed highest in animal caps (ectoderm) are classified as animal cap enriched genes (AC), genes expressed highest in marginal zone (mesoderm) explants are classified as marginal zone enriched genes (MZ), and genes expressed highest in vegetal mass (endoderm) explants are classified as vegetal mass enriched genes (VG). For instance, the expression level of foxi1 is 107 TPM in the animal cap, 10.6 TPM in the marginal zone, and 1.2 TPM in vegetal mass. Since Foxi1 expression is highest in AC, it is assigned as an animal cap enriched (AC) gene. This assignment is not perfect but practical.

We also examined whether the mis-regulation of the ectodermal genes in endoderm, or endodermal genes in ectoderm is statistically significant using Fisher’s exact test. For instance, the first bar in Figure 5F indicates that 6,170 genes among 28,850 annotated genes are endodermal genes (VG, colored in green), which is 21.4%. The second bar shows that 43 out of 92 genes upregulated after HDAC inhibition in ectoderm (AC) are endodermal genes (VG, green), which is 46.7%. Fisher’s exact test indicates that the difference is statistically significant (p-value < 6.1e-8). Similar analysis showed that ectodermal genes are preferentially upregulated in the endoderm. We suggest that HDAC activity is required to safeguard the misactivation of “inappropriate” germ layer genes in each region of the embryo.

I also think the authors should highlight and discuss that TSA treatment has a much more profound effect on gene expression dysregulation in the AC compared to the VG (Figure S5B). There are many more genes up and downregulated in the AC. This is quite a novel and interesting result. Can the authors comment on why?

We have performed HDAC inhibition analysis using two HDAC inhibitors, TSA and VPA, and noticed that TSA does induce many more genes in AC than VG cells when the data are compared to each other (Figure S5B). We speculate that this may be a TSA-specific effect because TSA is known to inhibit non-Class 1 HDACs. Therefore, we focused on the genes that are affected by both TSA and VPA. However, even with this analysis, we still found many more genes to be affected in AC than VG. Perhaps, this is due to the large-scale difference in timing of zygotic gene activation occurring earlier in AC than VG (Chen and Good, 2022, PMID: 36007528). Since this is largely a speculation, we simply report on the finding.

2. Timing of HDAC1 binding to the genome. It is interesting that HDAC1 peaks are widespread in the genome by Stage 8, presumably during zygotic genome activation, and that these peaks largely strengthen over time into the early gastrula. This however begs the question of when HDAC1 begins to bind to the genome. Is it already bound in the egg, or upon fertilization, or are peaks only detectable starting around the time of genome activation? Given that the protein is present in the egg and throughout cleavage stages (Figure 1A), and that the authors propose FoxH1 and Sox3 may be contributing to recruitment, the manuscript would be strengthened by characterization of the time of recruitment.

We performed a time-course ChIP-qPCR analysis of selected regions, representing 9 identified Hdac1 bound regions, and 2 negative control regions. Figure 1G suggests that Hdac1 does not bind to the genome prior to st8 (ZGA).

We also note that the number of Hdac1 binding sites at st8 is low (n=1,340) compared to those of Foxh1 (Charney et al., 2017, n=28,611), Sox3 (Gentsch et al., 2019, n=8,368), Vegt (Paraiso et al., 2019, n=21,711), and Foxi2 (unpublished, n=13,158) at the same stage. Additionally, Hdac1 binding strength also seems to be weak (Figure 1F) at st8. Based on this evidence, we suggest that the timing of genomic engagement of Hdac1 differs from that of the maternal transcription factors.

3. Differential role for HDAC1 and HDAC2 in lineage maintenance? The TSA treatment makes it difficult to disentangle specific subsets of genes modulated by promoter-proximal deacetylation. Can the authors contrast TSA vs. VPA inhibition experiments to identify peaks that could likely be attributed to HDAC2 rather than HDAC1, and perform GO analyses? Alternatively, can the authors provide direct binding data with ChIP-seq of HDAC2 in blastula and gastrula.

New Hdac2 ChIP-seq results are summarized in Figure S1E-H. Since we performed

Hdac2 ChIP-seq experiment once, the peak numbers of Hdac2 are greater than that of Hdac1, which underwent IDR analysis. Despite this, overall, Hdac2 binding is observed in the regions where Hdac1 binds. Importantly, Hdac2 binding is notably stronger around Hdac1-Hdac2 co-bound peaks than around Hdac2 lone peaks (Figure S1H). This finding is consistent with the previous findings that Hdac1 and Hdac2 can assemble into the same HDAC complexes (Seto and Yoshida, 2014, PMID: 24691964; Milazzo et al., 2020, PMID: 32429325).

We attempted Hdac1 MO knockdown (see Source Data), but were only able to reduce the Hdac1 protein level in st9 and st10.5 embryos, but not in st8 embryos, which is likely due to the timing of degradation of stores of maternally deposited Hdac1 protein (https://www.xenbase.org/entry/gene/geneExpressionChart.do?method=drawProtein&g eneId=865283&geneSymbol=hdac1.S&addProteins=hdac1) in the early embryo. Therefore, we are unable to address the specificity difference between Hdac1 and 2 at present time.

4. Direct evidence that HDAC1/2 binding contributes to gene expression regulation. The authors utilize inhibition of deacetylation (TSA) or HDAC1 specifically (VPA) to analyze changes to pan-H3K-acetylation and gene expression, regionally within the AC and VG. In general, I like this line of experiments, however, I wonder whether the TSA inhibition might be affecting other deacetylase enzymes, and what the effect would be of MO knockdown of the HDACs (or their recruitment via FoxH1, Sox3). Other groups have found that HAT-HDAC binding to promoters and enhancers does not always correlate to gene expression regulation. Extending this notion here, I worry that the authors may not be directly analyzing the downstream function of HDACs. Can the authors compare their differential expression data to existing gene profiling studies in embryos that have knockdown of FoxH1, Sox3, or HDAC1/2 to demonstrate similar profiles of dysregulation?

At present, Sox3 knockdown RNA-seq data is not available. Therefore, our analysis is focused on Foxh1 KD data. We compared the differentially expressed genes in Foxh1 knockdown and TSA-treated animal cap explants (Zhou et al., 2022, PMID: 35848281), and found that only a small subset of genes are co-regulated by Hdac1 (defined by TSA sensitive and Hdac1 genomic binding) and Foxh1. However, we found that Sox3 and Foxh1 double MO KD results on modest reduction (~2 to 3 fold on average) of Hdac1 binding (Figure 2F). Therefore, the recruitment of HDACs to individual genomic regions is likely to be complex, involving multiple maternal TFs. The result is not surprising as there could be multiple TFs that recruit Hdacs.

Reviewer #2 (Recommendations for the authors):As a general reader, I do not know enough of the background knowledge in the field to know how novel the observations are, and it seems that most of these specific comparisons are made in the discussion, and it is not explicit what observations are really new, or whether the importance of the paper is in the documentation. The authors need to draw this out more clearly. Also, as a general reader without the ingrained knowledge of the activating or inhibitory functions of the different histone modifications, I initially found it impossible to hold onto the threads of the arguments but was able to do so by color coding the modification types as activating, inhibitory, or heterochromatin in the text. I would hope that modern journals might be able to use colored text to enable readers to read and evaluate the text more readily. Similarly, it is hard to hold on to the nature of different classes of response, so I suggest more descriptive names for the classes. Without these, the manuscript is very tough going for the general reader. Indeed, as a detailed examination of chromatin modification, it is inevitably dense. Again as a general reader, I would find it more useful to present the specific genes that convey the arguments in the main text and figures, so that there is something to get one's teeth into and move the venn diagrams and stacked bar graphs and violin plots, which I find uninformative for the specific arguments, but which are necessary for the generality of the claims, to the supplement. But these are suggestions to make the manuscript more accessible to the general reader. If the target audience is the specialized field of chromatin organization or detailed regulation of gene expression in developing animals (without going beyond inhibitor studies to the sequences responsible for the DNA recruitment or the enzymatic activity of the protein), then the paper would probably be more appropriate for a more focused journal. But if there are new insights that can be more clearly articulated, then the manuscript could be appropriate to eLife.

We have incorporated the suggested changes.

Major points, in no particular order.1. While the Hdac genes are shown for mRNA it would be useful to see whether this matches the protein databases. Are they detectable, and does mRNA translate to protein abundance?

Currently, there is no database that would allow us to compare the protein levels between different HDACs directly in *Xenopus*. While mass spectrometry datasets (Peshkin et al., 2019; Nguyen et al., 2022) available in *Xenopus laevis* are quantitative, the data for each protein is relative abundance over developmental time points and doesn’t permit comparisons between the levels of different HADC proteins.

2. Line 140- were Vegt peaks found to be enriched in the maternal motifs? (Gentsch?)Indeed, was there enrichment of other motifs associated with transcriptional activation as found by Gentsch et al? (POU-SOX (Pou5f3-Sox3 heterodimer), Krüppel-like zinc finger (ZF; Sp1 and several Klf), POU (Pou5f3), SOX (Sox3), bZIP (Max), FOXH (Foxh1), ETS (Ets2), NFY (NFYa/b/c), SMAD (Smad1/2), VegT (mVegT)), A lack of enrichment would illustrate a strong difference in recruitment and be informative.

We now include a list of the top 50 motifs (ranked by FDR) in our Supplementary File 1 for all 3 stages. At st8, POU motifs are highly enriched. Among various FOX motifs, Foxh1 motif is the most enriched. Some POU-SOX, SOX motifs are also present. At st9, SOX, POU, POU-SOX, and FOX motifs are highly enriched. ZIC (including a few KLF) and HOX motifs are also detected, but ranked below the top 100 enriched motifs. T-box (Eomes) motifs are captured below the top 200. At st10, FOX motifs are rare.

Instead, ZIC motifs (including ZIC, KLF, and SP) are highly enriched. SMAD, SOX and T-box (Eomes) are present within the top 150 enriched motifs. ETS, MAX, and NFY motifs are present below the top 200. The appearance of new motifs such as ZIC, Tbox, and ETS, and the disappearance of POU and FOX motifs at the later developmental stages is consistent with the model that different TFs dynamically recruit Hdac1 to the genome.

3. "Consistent with the overlapping binding of Ep300, Hdac1 peaks display a moderate level of H3K9ac (Cluster c and d), high levels of 164 H3K18ac (Cluster b, c, and d), H3K27ac (Cluster b, c, and d), and pan-H3 lysine acetylation (pan-H3Kac)."Why is this consistent (ref required?) is Ep300 always an activating signal? The next section discusses activating signals, so maybe so, but it does not say so. What is the current understanding of Ep300? General enhancer binding?

Ep300 is a histone acetyltransferase (HAT) that catalyzes histone acetylation at H3K9, K18, K27 (Zhang et al., 2018, PMID: 30150647). Therefore, the presence of Ep300 is consistent with the presence of histone acetylation. In general, Ep300 is highly correlated with active enhancers, and so is histone acetylation (Heintzman et al., 2007, PMID: 17277777; Ong and Corces, 2011, PMID: 21358745). We clarified this point in the text (p9).

I fear that the paper is exceedingly hard to follow unless one has these identities imprinted in one's own brain, which I don't. Indeed, for people not in the immediate field, the nomenclature, with its acetylations, and multiple methylations is all too easy to forget. For ease of reviewing, I color-coded the histone marks according to Wikipedia. It made the manuscript much easier to follow, though I am sure my simplistic green or red color code is too simple. The color coding of the figures in the manuscript is more nuanced, and presumably reflects some deeper understanding. It would be useful to share this with the reader. Otherwise, the activating (green) repressive (red), and heterochromatin (brown, blue? - what are these?) and the main text should also match an informative color coding, which should be trivial in these days of digital typesetting.

Thank you for your suggestion. We added descriptive words, such as “activating” and “repressing” throughout the text to improve the clarity of the text. The ChIP-seq data under a specific color panel (e.g., Figure 3) represents specific ChIP-seq data sets (e.g., H3K9ac, H3K27ac, etc) from different developmental stages. This makes it easier to compare the ChIP-seq results across different experiments.

4. "we speculate that these regions are differentially marked in space due to heterogeneous cell populations present in the whole embryo. Hdac1 Cluster I peaks are referred to as heterogeneous CRMs."Surely the authors can give explicit examples here for some known heterogeneously expressed genes to reduce the speculative nature of this group? (though they do eventually show up in the supplements- but why not progress the argument from the specific to the general?)

We now provide specific examples in the text as we discuss the data. We thank the viewer for this suggestion as this change has improved the clarity of the manuscript.

5. Ideally, the figures and legends should be interpretable independent of the text, for example, when wondering what abcd are in figures 1 and 3, one should not have to search in the text for the relevant explanation

Heatmaps in Figure 1C and Figure 3A and B, all include ChIP-seq data of Hdac1 segregated into 4 clusters (a, b, c and d). This was intended to reference all other relevant data against that of Hdac1 (a, b, c and d) clusters. We now more explicitly explain these panels in the text, figures, figure legends.

"Overall, a minority of Hdac1 peaks are unique to each of the blastula stages (Cluster a and b) while a majority of Hdac1 peaks are present across multiple stages (Cluster c) and at the early gastrula stage (Cluster d) (Figure 1C, S1D). "And in reading this, "unique to each blastula stage", could be expressed more clearly to indicate their uniqueness- ie at stage 8 (cluster a, likely driven by maternal factors?) and stage 9, (cluster b, likely driven by newly expressed zygotic factors or signals?).Re the italics – where are the genes known to be in these classes? Ie the Nodals, vs genes driven by zygotically activated signals like Smad2? It is important for a simple argument to give the general reader some specific biological context.

We now added the temporal expression profiles of genes associated with Hdac1 binding (Figure 1D). We focused our analysis on the expression of zygotically expressed genes (TPM <1 between 0 and 1 hour post-fertilization, and TPM >1 at any time in the window of 1- 23 hours post-fertilization), and excluded maternally expressed genes from this analysis.

In general, the timing of Hdac1 binding is correlated with the activation of genes. Cluster a (peaks uniquely present at st8) associated genes are activated at 4hpf (p=0.0035). Cluster b (peaks uniquely present at st9) associated genes are activated at 5hpf (p=8.56e-5). Cluster d (peaks uniquely present at st10.5) associated genes are also activated at 5hpf (p<2.2e-16). Interestingly, cluster c (peaks that are present in st8 and/or st9 and/or 10.5) associated genes exhibit activation at 4hpf (p=5.93e-8). In terms of gene expression levels, cluster a genes are least activated whereas cluster c genes remain activated at the highest level after 5hpf. The expression levels of cluster d

genes gradually exceed that of cluster b as development proceeds. We have briefly summarized these findings in the text p6.

6. Likewise "These observations suggest that Hdac1-mediated suppression is largely dictated by developmental programs" – perhaps the authors might give some examples in the text.

We modified this sentence and state, “Hdac1 binds to facultative heterochromatic regions facilitating the repression of genes”. This modification is needed as we do not have a molecular understanding of how it works and therefore decided the above change would be appropriate.

7. "First, an increase of pan-H3Kac is observed across Hdac1-bound CRMs, including active CRMs, consistent with the canonical enzymatic activity of Hdac1" remind us what H3 Kac is for – it seems to be in the purple, brown and red categories. Is it associated with transcription start sites? It's not clear what antibody was used. It seems it must be a pan H3Kac antibody?

We now state in the text that pan-H3Kac antibody detects histone 3 K9, K14, K18, K27, and K56 acetylation. Regarding the question about H3Kac being associated with open chromatin that is transcriptionally active, we cannot conclude that all these acetylation are associated with transcription start sites. This is because we have used pan-H3Kac antibody, which recognizes various forms of acetylation on H3.

8. What are the effects of Hdac inhibitors on the transcriptome? Seems that this is an essential piece of information to interpret the results on chromatin? Seems there is a lot of literature on this, and so far as I can tell it is not cited until the discussion. It would be nice to have a digested understanding to compare to the chromatin results in the Results section (some of this appears eventually, but late enough to leave the reader frustrated when the topic comes up).

We stated on p4, ln 75-77, that HDAC inhibition in cell lines results in both up- and down-regulation of genes, with several papers cited.” We have also modified sentences (p4, ln 75-80) to better describe these experiments.

9. I had great difficulty understanding the paragraph that begins "Given that Hdac1 CRM clusters (Clusters I-IV in Figure 3C) are subjected to both active and repressive epigenetic modifications presumably in different germ layers, we compared the general status of H3 acetylome (pan-H3Kac) between two distinct germ layers" I think that all of this is VPA and TSA independent information? But in the end, isn't it all circular? Ie higher enrichment of pan H3Kac associated with higher expression?

In Figure 3C, we focus on H3K27 modification. Lysine 27 can be either acetylated or methylated, but cannot accommodate both modifications simultaneously on the same histone polypeptide. However, we found a cluster (cluster I) of regions marked with both modifications. For instance, Figure S3C shows that the surrounding regions of ectodermally expressed lhx5 (Houston and Wylie, 2003, PMID: 12736213) are marked with both active H3K27ac and repressive H3K27me3. Our hypothesis is that lhx5 is marked by active H3K27ac in the ectoderm cells, but lhx5 in endoderm cells is marked by repressive H3K27me3. This ensures that lhx5 is specifically expressed in ectoderm, but not in endoderm. This hypothesis is confirmed by the analysis of active pan-H3Kac data using dissected AC and VG explants (Figure S4.2F). CRMs of lhx5, kcdt15, and foxi1 are acetylated (pan-H3Kac) in AC tissue, but not in VG tissue. We also find foxa4, mixer, sox17a, vegt are acetylated in VG, but not in AC. We have modified p12 and added a new sentence (ln 248-251) and a figure (FigS4.2C) to clarify the statement.

10. "Taken together, these data demonstrate that Hdac1 maintains differential H3 acetylation states between germ layers through its catalytic activity." Is a rather bland statement after trudging through the paragraph. I think that one can make the general statement that upon inhibition of HDAC, repressed genes have a larger increase in their pan H3Kac than do active genes. And is there evidence that it is through the catalytic activity rather than its binding?

Thank you for the suggestion, and we have modified the text (see p14, ln815-816). In addition, we investigated the binding of Hdac1 upon TSA inhibition. Figure S4.1C and

D show that the genome binding ability of Hdac1 is largely insensitive to TSA treatment.

Hence, the changes of histone acetylation patterns is due to the loss of the enzymatic activity of HDAC and not the result of binding alone.

"we found that the fold increases of pan-H3Kac signals after TSA treatment are negatively correlated with the levels of endogenous pan-H3Kac signals in both AC and VG explants" This is an unnecessarily complicated sentence. It's already hard to figure out the direction things are going in without having to think about what negative correlation means in these contexts. But I think it means what I just suggested?

We have modified the sentence and it now states,

“Interestingly, CRMs with low endogenous levels of histone acetylation tend to be more responsive to HDAC inhibition, than CRMs with high acetylation” (p 13, ln 607-608).

11. I ask that the authors consider how much time would be spent by the reader looking back to see what Class1,2 and 3 genes are. Could they not be more readily described as "selectively activated", "constitutively repressed" and "Constitutively activated" if these were designated SA, CR, and CA that might be easier to remember/interpret, especially for me who can't remember class I and 2 from one paragraph to the next. Or consider the value of spelling them out each time, at least for the reviewer. Once accepted, one can return to uninformative abbreviations and let the reader suffer.

Our original intention was to provide the reader the opportunity to interpret the data without our preferences. However, we realize the difficulty of keeping track of these uninformative names. We now added a short description for each gene class when it is helpful.

12. What is the mechanism of Hdac inhibition by VPA and TSA? Does the Hdac dissociate from chromatin, allowing access to acetylases? Or does it stay on the chromatin in an inactive form, suggesting a more complex mechanism of the differential effects of inactivation? What do the data say about the potential mechanism?

TSA chelates the Zn2+ cation in the active site of histone deacetylase enzymes (Finnin et al., 1999, PMID: 10490031) and inhibits the enzymatic activity. The mechanism by which VPA inhibits HDAC activity still remains obscure. VPA has been shown to block the interaction between Sp1/Sp3 zinc finger TFs and Hdac1/2 (Davie 2003, PMID: 12840228), and to induce ubiquitin-mediated Hdac2 proteolytic degradation (Kramer et al., 2003, PMID: 12840003). ChIP-seq analysis of Hdac1 after TSA inhibition suggests that TSA generally does not interfere with its association to the genome (Figure S4.1).

Reviewer #3 (Recommendations for the authors):Figure 11C: St101/2 or stage 9-specific HDAC1 peaks seem to already show signal at stage 8. (Difficult to grasp relative level as no indication of what the blue scale is….). In any case, is this low level at stage 8 meaningful (how does it look in all enhancer peaks for example).

We have now assigned scale bars to all heatmaps. At present, we are unable to conclude whether the weak signals (clusters b and d at stage 8) are meaningful as the peaks were not statistically significant. However, we note that these weakly bound regions gradually gain binding strength over time. We, therefore, described the binding dynamic of Hdac1 to be progressive (page 7, ln 162).

What is the temporal expression (early blastula to neurula stages for example) of the a-d set of HDAC1-associated genes? Any correlation between stage-specific binding and expression pattern would strengthen the proposed involvement of HDAC1 in the epigenetic regulation of genes around ZGA.

Figure 1D describes the correlation between the Hdac1 binding and the temporal expression profiles of Hdac1-associated zygotic genes in cluster a-d. In general, the timing of Hdac1 binding correlates with the activation of associated genes. Genes associated with cluster a are activated first (4 hpf), and then cluster b and d associated genes (5 hfp). Cluster c associated genes are typically activated between 4-5 hpf.

Figure 2:2B What are the enriched motifs in stage 8 and stage 10 specific peaks?

We now include a list of the top 50 motifs (ranked by p-value) found in stages 8, 9 and 10.5 (see the source data).

2C: How does the HDAC1 peak intensity look if you integrate the HDAC1 chip a-amanitin data in the map from 1C (in complement to S2B).

Pearson correlation analysis comparing Hdac1 peak intensity between uninjected control and a-amanitin injected embryo shows that there is no significant difference between these samples (Figure 2A).

Charney et al. document binding of TF pre/post ZGA.To support the ID that TF binding is associated with HDAC1 binding it would be important to check if stage-specific binding of HDAC1 is associated with stage-specific binding of TF.

Figure S2I shows heatmaps of the Hdac1 peaks at stage 8, 9 and 10.5 with Foxh1 and

Sox3 peaks. Hdac1 peaks overlap well with Foxh1 and Sox3 ChIP-seq peaks. The

Foxh1 and Sox3 peaks at cluster b, c and d precede that of Hdac1. For instance, while Foxh1 and Sox3 peaks are present in stage 8 and 9 samples, Hdac1 speaks appear at stage 9. Similarly, Foxh1 and Sox3 peaks in cluster c are present in stage 8, Hdac1 peaks appear at stage 9 and persist. Lastly, the peaks of Hdac1 at stage 10.5 appears after the appearance of Foxh1 and Sox3 peaks at stage 9. These data is consistent with the view that Hdac1 binding occurs after the appearance of Foxh1 and Sox3 binding.

The title "HDAC1 binding is instructed maternally" is not appropriate. Interference with maternal TFs would be needed to go beyond correlation. HDAC1 binding after foxh1MO (as in Charney et al) will indicate if HDAc1 binding depends on the presence of this maternal TF.

Here we provide two pieces of evidence that support the statement. First, inhibition of RNA pol II activity by a-amanitin proves that Hdac1 binding to the genome at blastula stage is independent of zygotic factors (Figure 2A, Figure S2A-C), and therefore its recruitment to the DNA must be instructed by factors supplied in the egg/embryo maternally. Based on this evidence alone, we think the current heading of this section is appropriate. Second, Hdac1 binding after Foxh1 and/or Sox3 MO injection is reduced in foxh1 and sox3 morphants (Figure 1F). The partial knockdown of Hdac1 recruitment in the absence of maternal Foxh1 and Sox3 is consistent with the model but additional TFs are likely involved in its genomic recruitment.

Figure 3"moderate" level of K9ac etc….moderate compared to what??? I would think comparing for example level of p300+ HDAC+ to p300+HDAc- would give an actual idea if the level on HDAC associated p300 peak stands out.The same goes for figure S3B.Without such kind of comparison, it is difficult to conclude that HDAC1 peaks have biased epigenetic status compared to all p300 peaks for example.Again in Figure 3B HDAc1 bound region exhibits a strong signal density of K27me3.

We agree with the referee’s comments and have removed quantitative words in our description and simply state the presence of epigenetic marks at Hdac1 binding sites.

Line 183: I don't think cluster b fits the description.

We modified the sentence. Instead of stating, “Hdac1 peaks (Cluster b, c, d, but not a) are marked by both active and repressive epigenetic signatures…”, we changed the sentence to “a majority of Hdac1 peaks (cluster c and d) are marked by both active and repressive epigenetic signatures…”

Fig3CD why exclude HDAC peaks from stage 8?

Since these peaks are free of notable epigenetic signatures (e.g., Ep300, RNA pol2, and histone acetylations and methylations), it is unclear what the physiological relevance/function of these peak regions during germ layer specification. Therefore, we excluded them from the current analysis.

It would be useful to see how HDAC peaks signal looks on Figure 3D map.

We now include Hdac1 peak data (st9 and st10.5) in Figure 3D. We have also modified the text to better describe the behavior of cluster 1.

In 3D there seems to be a clear anti-correlation between K27me3 and K27ac signal (see cluster I) that does not really fit the description of line187-191. This is important because the concept proposed in the manuscript relies on these co-occurring K27ac/me sites. Are the signal anti-correlated in cluster I?

Regarding the deposition patterns and density of H3K27me3 and H3K27ac being different, we offer the following explanation. First, H3K27ac peaks are called using a NarrowPeak format, which identifies narrow sharp peaks resembling those of transcription factors. On the other hand, H3K27me3 marks are found using a BroadPeak format because methylation signal on H3K27 is generally broader (Zhou et al., 2011, PMID: 21116306). The difference in peak calling will impact both the peak position and the overall peak intensity of H3K27ac and H3K27me3. This may result in peaks that look different in the heatmap. However, when these regions are examined (Figure S3C), the overlapping regions are readily visible (see response below).

S3C K27ac and me do not seem to overlap much. Can we see the region where peaks are detected (add a box where peaks are detected)?

Figure S3C shows the gene browser view of various regions (specific peak overlaps are now boxed). There are significant peak overlaps between H3K27ac and H3K27me3 around lhx5 (Class I), but not in Class II, III and IV genes.

Line 203 – before testing of this hypothesis is surely again to compare acetylation in p300+ HDAC+ versus p300+ HDAC- (or at TSS) to evaluate if TSA primarily affects HDAc1 sites?

To demonstrate that Hdac1 bound regions are preferentially affected by TSA, we randomly sampled 23,442 genomic regions (an equivalent Hdac1 peak number) and compared the changes of pan-H3Kac in these randomized regions vs. Hdac1 peaks. First, we observed that the change of pan-H3Kac is lower in randomized regions than that of Hdac1 peaks (Figure 4E, S4.3A). We next examined the levels of pan H3Kac across clusters (Figure S4.3E, S4.3F) as well as fold changes of pan-H3Kac after TSA treatment (Figure 4F, S4.3D). TSA affected Hdac1 bound regions more than randomized genomic regions in both cases. Based on these analyses, we conclude that TSA preferentially targets Hdac1-bound regions to alter the levels of pan-H3Kac.

Figure 4A/BCan the author use a TSA control experiment where TSA is applied only prior to ZGA or only around ZGA?? This will indicate what is the time window when TSA has this effect.It would be also important to document the effect on the development of whole embryos of the TSA treatment scheme used on explant (Figure 4C). Are the embryos defective in these conditions?

The suggested experiment is difficult to conduct because we do not know how quickly (or how deeply) HDAC inhibitors diffuse into *Xenopus embryos* and can be washed away after the treatment. Since the *Xenopus tropicalis* embryos progress from stage 8 to 10.5 in about 3 hours, it would be very challenging to perform treat-and-wash-away experiments in such a tight developmental time window.

4C:What is the effect of TSA on the number of peaks detected?Would be very important to have a heat map that summarizes the panAc-peak in control and TSA conditions as well as the HDAc1 peak. This would illustrate to which extent TSA primarily affects HDAC binding sites.

Based on the TSA-induced global hyperacetylation (shown by westerns in Figure 4B, and quantitative ChIP-seq in Figure 4E, S4.3A), we expect to detect more peaks in TSAtreated samples than in DMSO-treated samples. Indeed, spike-in normalized peak calling revealed that TSA treatment not only induced new peaks in AC or VG explants (Figure S4.3B, S4.3C, cluster b), but also elevated common pan-H3Kac signal (a).

AC and VG-specific peaks seem to exhibit a much lower intensity of panAc signal than common (scale missing on the map…). What could be the reason for such a difference?

We are a bit confused with this question. If this referee refers to the intensity difference, yes he/she is correct to notice that. Consistently, the common cluster (cluster A) displayed a stronger H3K9ac and H3K27ac signal density than the germ layer specific clusters B and C. We also found that genes associated with cluster A are expressed significantly higher (p< 2.2e-16) than genes associated with germ layer specific clusters B and C, shown in Figure S4.2C. This is consistent with the observation that histone acetylation levels generally positively correlate with gene expression levels. We briefly state this observation in the text (p 12, line 555). Perhaps, the genes belonging to cluster A have higher acetylation density because these genes need to be expressed at high levels. A scale bar is now added to Figure 4C. Alternatively, if the referee is asking why the AC and VG intensities, presumably in cluster A, are significantly weaker that in whole embryos, we do not know the reason why this is the case. Additionally, this is not a major point of this paper, and we prefer not to comment on this issue in the paper.

4E FC I assume is FC TSA / control.

You are correct, and the heading of Fig4F is better labeled now.

Fig4E and S4GGlobal increase in acetylation in embryos (4D) – immediate question whether acetylation on the genome disproportionately affects HDAC1 binding sites?This is critical since the authors assume that TSA effect indeed reflects the effect on HDAC1 activity. Authors, unfortunately, focus analysis solely on HDAc1 binding sites so the conclusion line249 is not yet supported.

We performed Hdac1 ChIP-seq in TSA-treated embryos at st9 and st10.5 and found that TSA treatment has little effect on the Hdac1 genomic binding profiles (Figure S4.1). Therefore, the elevated acetylation by TSA does not alter the DNA binding ability of Hdac1.

All observations could also fit the model whereby TSA leads to the homogeneous increase of ac level genome-wide – would obviously lead to lower fold change if starting point high (cluster III) and higher if low (cluster II).

Our data do not support a model of homogeneous increase. We examined a fold change as shown in Figure 4F and Figure S4.3D. CRMs of Hdac1 clusters I-IV respond differentially upon HDAC inhibition in both animal and vegetal explants. We also plotted the difference (subtraction) of normalized counts of pan-H3Kac (Figure S4.3E and S4.3F). Compared to randomized genomic regions, Hdac1 all peaks displayed a much higher level of differences in pan-H3Kac, suggesting that TSA inhibition does not lead to a homogenous increase of histone acetylation level genome-wide; instead, it had a more profound effect at genomic regions bound by Hdac1.

Figure 5When excluding genes with maternal transcripts how many genes are left in each category?

We added the numbers of genes examined in both Figures 5B and 5C.

How do sets of genes with the same epigenetic configuration but HDAC- compare (i.e genes that are K27me3/K27ac but without HDAC1?).

This is an interesting question, but somewhat difficult to execute because a large number of genes bear multiple CRMs, of which each CRMs can have different epigenetic signatures. Therefore, we first excluded genes that have multiple CRMs bearing various epigenetic signatures (e.g., H3K27me3 and H3K27ac), and examined the expression of the remaining genes. Class I and III genes that are bound by Hdac1 are preferentially expressed at the blastula and gastrula stage, compared to the class I’ and III’ genes that are devoid of Hdac1 peaks (Figure S3E, 4-8 hpf).

VPA analysis cannot support the claim that TSA treatment has no off-target effect because this (i) is not RNA-seq analysis and (ii) 8/24 genes selected for RTqPCR are not showing the expected effect.

We now include RNA-seq results of VPA-treated embryos. Overall, a large number of genes affected by TSA are also affected by VPA (Figure S5A). When we compared the expression levels of genes affected by TSA alone, VPA alone, or both VPA and TSA, similar trends of gene expression changes are observed (Figure S5B). For the spatial gene expression change analysis upon HDAC inhibition, we focused on genes coaffected by TSA and VPA to minimize non-specific effects.

The first important question to address: are the TSA-affected genes disproportionally enriched for HDAC peaks associated genes?

In Figure S5D, we show the frequency of genes that are affected by TSA alone, VPA alone, and both VPA and TSA treatments. Overall, a majority of genes (>50%) affected by Hdac1 inhibition have functional Hdac1 binding site nearby. HDAC inhibitors had the most impact on the genes that are affected by both VPA and TSA.

5E Not sure to get the significance of observation.Can it be that a gene can be detected as upregulated only if they have a low level in control conditions can be detected as downregulated only if they have a high level in control conditions? Which is the rationale for the selection of Ac or V-specific genes…?

In this figure (now Figure 5F), we aim to address whether the expression of genes differentially regulated by TSA shows spatially restricted expression patterns. For example, are endodermally expressed genes preferentially upregulated in ectodermal lineage cells upon TSA treatment? To tackle this question, three components are needed: a list of genes classified as upregulated in ectodermal cells upon TSA treatment (Up in AC in Figure 5F), a list of genes expressed in endoderm (labeled as VG) in normal embryos (based on Blitz et al., 2017), a list of genes that are preferentially expressed in AC or VG (see the description of statistical treatment below).

The method of assigning genes to spatial expression patterns was published previously

(Paraiso et. al 2019; PMID: 31167141) and also described in Materials and methods- Categorical Analyses. Briefly, genes expressed less than 1 TPM are categorized as not expressed (NE), genes whose coefficient of variation is less than 10% among 3 germ layers are considered not locally expressed (NL), genes whose expression is highest in animal cap (ectoderm) explants are classified as animal cap enriched genes (AC), genes whose expression is highest in marginal zone (mesoderm) explants are classified as marginal zone enriched genes (MZ), and genes whose expression is highest in vegetal mass (endoderm) explants are classified as vegetal mass enriched genes (VG). For instance, the expression level of foxi1 is 107 TPM in the animal cap, 10.6 TPM in the marginal zone, and 1.2 TPM in vegetal mass. Hence, foxi1 is an animal cap enriched (AC) gene (and is in the salmon red area in Figure 5F). This assignment is not perfect, but practical as foxi1 is a known regulator for ectoderm specification (MatsuoTakasaki et al., 2005, PMID: 16079156; Mir et al., 2007, PMID: 17229765).

We applied Fisher’s exact test to examine the proportional significance between two groups of genes. For instance, the first bar (All genes) of Figure 5F shows among 28,850 annotated genes, 6,170 genes (VG, 21.4%) are endodermally expressed genes at this stage of development. The second bar shows among 155 genes that are upregulated after TSA treatment, 71 genes (VG, 45.8%) are endodermal specific genes (VG). Fisher’s exact test shows that the enrichment of endodermal gene upregulation in anima cap (ectodermal cells), that is 45.8% (test group), is statistically significant over that from all annotated genes (21.4%, reference group), resulting in a p-value of 1.32e11. We conclude that endodermal genes are preferentially upregulated in the animal cap cells upon HDAC inhibition, and the data is consistent with the view that HDAC activity is required to safeguard misactivation of spatially restricted genes.